# 3D optogenetic control of arteriole diameter in vivo

**Philip J O'Herron[1,2]\*, David A Hartmann[2,3], Kun Xie[1], Prakash Kara[2,4,5], Andy Y Shih[2,6,7,8]**

[1]Department of Physiology, Augusta University, Augusta, United States; [2]Department of Neuroscience, Medical University of South Carolina, Charleston, United States; [3]Department of Neurology & Neurological Sciences, Stanford University, Stanford, United States; [4]Department of Neuroscience, University of Minnesota, Minneapolis, United States; [5]Center for Magnetic Resonance Research, University of Minnesota, Minneapolis, United States; [6]Center for Developmental Biology and Regenerative Medicine, Seattle Children's Research Institute, Seattle, United States; [7]Department of Bioengineering, University of Washington, Seattle, United States; [8]Department of Pediatrics, University of Washington, Seattle, United States

**Abstract** Modulation of brain arteriole diameter is critical for maintaining cerebral blood pressure and controlling regional hyperemia during neural activity. However, studies of hemodynamic function in health and disease have lacked a method to control arteriole diameter independently with high spatiotemporal resolution. Here, we describe an all-optical approach to manipulate and monitor brain arteriole contractility in mice in three dimensions using combined in vivo two-photon optogenetics and imaging. The expression of the red-shifted excitatory opsin, ReaChR, in vascular smooth muscle cells enabled rapid and repeated vasoconstriction controlled by brief light pulses. Two-photon activation of ReaChR using a spatial light modulator produced highly localized constrictions when targeted to individual arterioles within the neocortex. We demonstrate the utility of this method for examining arteriole contractile dynamics and creating transient focal blood flow reductions. Additionally, we show that optogenetic constriction can be used to reshape vasodilatory responses to sensory stimulation, providing a valuable tool to dissociate blood flow changes from neural activity.

**\*For correspondence:**
poherron@augusta.edu

**Competing interest:** The authors declare that no competing interests exist.

## Editor's evaluation

This paper will likely be of keen interest to researchers investigating vasculo-neuronal coupling – a proposed signaling mode opposite that of the more widely studied neuro-vascular coupling process. The optogenetics method described, inspired by methodology developed for interrogating ensembles of neurons, effectively enables simultaneous manipulation and monitoring of brain arteriole contractility in three dimensions.

## Introduction

The brain is a metabolically demanding organ, consuming 20% of the body's energy supply despite being only 2% of its weight (*Clark and Sokoloff, 1999*; *Raichle and Gusnard, 2002*). To ensure an adequate supply of oxygen and nutrients, brain arterioles dilate transiently in response to local increases in neural activity – a process called functional hyperemia (*Iadecola, 2017*). These vascular dynamics underlie hemodynamic imaging techniques used to map regional neural activity, such as blood oxygenation level-dependent functional magnetic resonance imaging (BOLD fMRI) (*Logothetis*

*and Wandell, 2004*; *Vanzetta et al., 2014*; *Raichle and Mintun, 2006*; *Howarth et al., 2021*). Arteriole dynamics are also crucial in buffering the brain from fluctuations in systemic blood pressure (autoregulation), and in driving clearance of metabolic waste products from the brain along perivascular spaces (*Cipolla, 2009*; *Zhao et al., 2015*; *Iadecola, 2017*; *Presa et al., 2020*; *van Veluw et al., 2020*; *Rasmussen et al., 2021*). Proper regulation of arteriole dynamics is essential to healthy brain function, as deterioration of neurovascular coupling, cerebral perfusion, and brain clearance contributes to the pathogenesis of many neurologic disorders (*Gorelick et al., 2011*; *Iadecola, 2013*; *Snyder et al., 2015*; *Sweeney et al., 2018*; *Rasmussen et al., 2021*). Thus, there is widespread interest in understanding vascular dynamics and how neural activity interacts with and shapes hemodynamic responses.

In vivo studies of functional hyperemia typically engage vascular dynamics indirectly by activating neurons (e.g., with sensory stimulation), which in turn leads to the hemodynamic response. This makes it difficult to study intrinsic properties of the blood vessels independent from neuronal activity and neurovascular coupling pathways. For instance, impaired vascular responses in disease may be due to reduced neural activity, altered neurovascular coupling, or loss of compliance in the vascular wall. Similarly, the kinetics of arterial contraction and relaxation in vivo cannot be understood independently of the kinetics of the neurovascular coupling mechanisms. The need to modulate arteriole diameter and/or blood flow independently from neural activation has led to the use of pharmacological approaches or systemic manipulations (such as a $CO_2$ challenge or modulation of systemic blood pressure) (*Kara and Friedlander, 1999*; *Leithner et al., 2010*; *Tarantini et al., 2015*; *Masamoto and Vazquez, 2018*). However, these methods do not provide spatiotemporal specificity for microvessels in the brain. Intracortical injection of vasoactive substances improves spatial specificity (*Cai et al., 2018*), but remains low throughput and can act on nonvascular cell types. Due to these limitations, there is a need for methods to directly activate the microvasculature with high spatiotemporal resolution.

Recently, optogenetic techniques have been developed to control vessel diameter with light by expressing opsins in smooth muscle cells. Most studies to date have used wide-field activation of ChR2 with visible wavelength light. However, this technique affects vessels over a large region and predominantly activates the superficial vasculature, limiting the spatial resolution achieved (*Wu et al., 2015*; *Zhang et al., 2015*; *Rorsman et al., 2018*). Some studies have improved the spatial resolution of optogenetics by using visible wavelength lasers to create a focal point of activation, but this also activates primarily surface vessels (*Hill et al., 2015*; *Mateo et al., 2017*; *Nelson et al., 2020*). Achieving greater imaging depths has been attempted with fiber bundle probes (*Kim et al., 2017*) but this reduces imaging resolution and causes tissue damage. Two-photon optogenetic activation allows deeper imaging and more precise spatial control than single-photon activation. This method has been used to activate ChR2 in mural cells to examine contractility with single-vessel resolution (*Hill et al., 2015*; *Tong et al., 2020a*; *Hartmann et al., 2021*). However, imaging and excitation in these studies required use of the same laser, limiting the ability to separate regions of stimulation from observation. Further, these studies used the original ChR2 (H134R) variant, and opsins with improved two-photon cross-sections have since become available.

Here, we introduce an approach to constrict cerebral arterioles with high spatiotemporal resolution up to hundreds of microns below the surface. We use the red-shifted opsin ReaChR (*Lin et al., 2013*) which has strong photocurrents and a long activation wavelength well suited for two-photon stimulation. This method allows for constriction of individual or multiple branches of pial or penetrating arterioles by driving cell-specific depolarization of smooth muscle cells. By combining dual-light paths and independent focusing of the excitation and imaging lasers, we constricted vessels independently from the depth of the imaging plane, making this a useful tool to manipulate and monitor vessel diameter in three dimensions. This approach is advantageous over ChR2 activation of vascular smooth muscle because it can manipulate single vessels over larger cortical depths, and longer wavelengths of light are less likely to create spurious vascular changes independent of opsin expression (*Rungta et al., 2017*).

## Results

ReaChR potently depolarizes cells when illuminated with red light (*Lin et al., 2013*), and exhibits peak two-photon activation efficiency at ~1000 nm (*Chaigneau et al., 2016*). This makes it ideal for combined two-photon activation and imaging of cerebrovasculature, similar to the all-optical

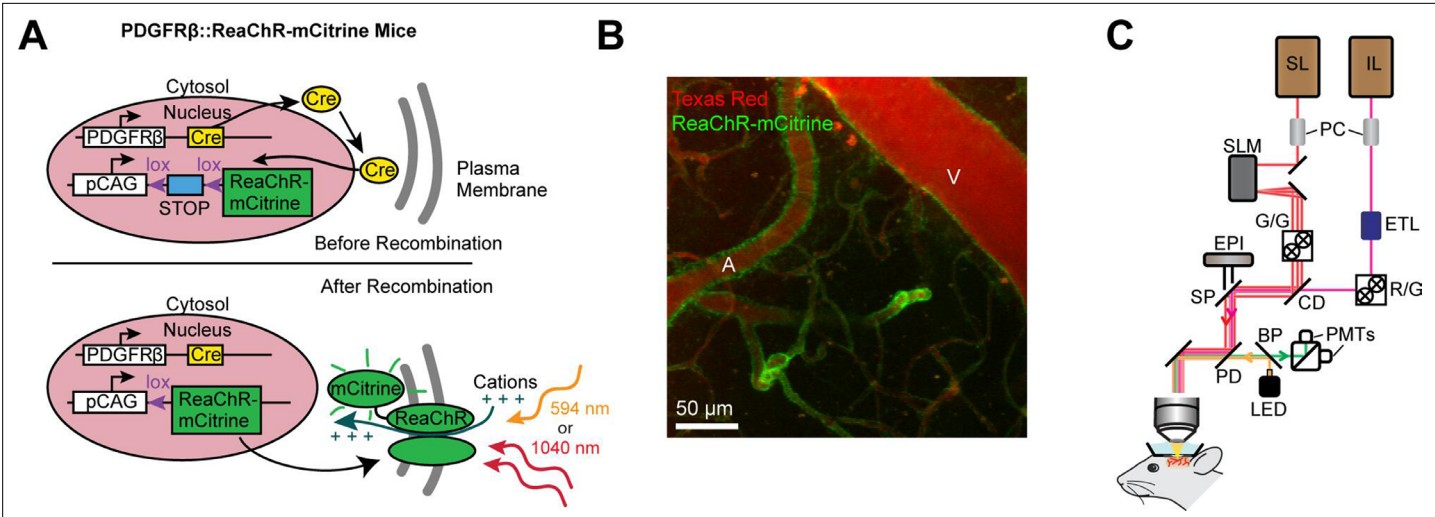

**Figure 1.** Mouse genetics and equipment for controlling arteriolar diameter with ReaChR. (**A**) Crossing the PDGFRβ-Cre line with a floxed ReaChR (tagged with mCitrine) line leads to expression of the opsin in vascular mural cells. Mural cells can then be depolarized with single-photon excitation using a 594 nm LED or two-photon excitation with a fixed 1040-nm pulsed laser. Studies with control mice (see *Figure 5—figure supplement 1*) followed a similar strategy except only YFP was expressed in the cytosol of mural cells. (**B**) Maximal projection image of cortical vessels from the mouse line created in panel A. The vascular lumen is labeled with Texas Red-dextran and the green in the vessel walls shows the expression of ReaChR-mCitrine. A = pial arteriole, V = pial venule. (**C**) Imaging equipment – the Ultima 2P-Plus from Bruker (see Methods for details). Imaging laser (IL), stimulation laser (SL), Pockel cells (PC), spatial light modulator (SLM), galvonometer mirrors (G/G), electro-tunable lens (ETL), resonant scanning/galvonometer mirrors (R/G), combining dichroic (CD), primary dichroic (PD), light-emitting diode (LED), short-pass mirror (SP), band-pass mirror (BP), photo-multiplier tubes (PMTs), epifluorescence module (EPI).

methods developed to activate neurons in vivo using long-wavelength light (~1040 nm) while simultaneously imaging with shorter wavelengths (~800–950 nm) (*Packer et al., 2015*; *Carrillo-Reid et al., 2016*; *Shemesh et al., 2017*; *Mardinly et al., 2018*; *Marshel et al., 2019*). To selectively express ReaChR in mural cells, we crossed floxed ReaChR reporter mice with PDGFRβ-Cre mice (see Methods; *Figure 1A*), which allows expression of ReaChR tagged with mCitrine in vascular smooth muscle cells and pericytes, with negligible off-target expression in neurons or astrocytes (*Hartmann et al., 2015*; *Figure 1B*). We imaged vasoconstrictive responses with two-photon microscopy while depolarizing the vascular mural cells either with a pulsed 1040 nm laser for two-photon activation, or a 594 nm light-emitting diode (LED) for single-photon activation (*Figure 1C*).

## Single-photon activation of ReaChR in vessel walls produces widespread vasoconstriction

We first examined ReaChR activation with single-photon stimulation using a 594 nm LED to stimulate vessels broadly across the cranial window. We found that a single 100 ms pulse of light led to the rapid and widespread constriction of pial arterioles, which then relaxed back to baseline values within 30 s of activation (*Figure 2A–D*; *Video 1*). We also observed a weaker constriction (~2–3%) in the pial venules (*Figure 2A–D*). Although some expression of the opsin was seen in venous mural cells (*Figure 1B*), the venous response was likely a passive deflation due to widespread reduction in blood flow caused by arteriole constriction given its slower time-course relative to arterioles (*Masamoto and Vazquez, 2018*).

We also observed that penetrating arterioles at least 300 μm deep in the cortex constricted with single-photon activation (*Figure 2E–G*; *Video 2*). When we applied repeated trains of 100 ms pulses, we were able to sustain penetrating arteriole constriction at 10–20% below baseline levels for over a minute while simultaneously imaging in between each pulse train (*Figure 2H, I*; *Video 3*). To demonstrate that constricted arterioles reduced downstream microvascular flow, we performed a line scan on a penetrating arteriole and the first branch of the arteriole–capillary transition (ACT) zone (85 μm deep in the tissue), to gather a measurement of blood flow to the capillary bed (*Figure 3*). The number of red blood cells (seen as dark streaks in the line scan) was greatly reduced in the branch within 1 s

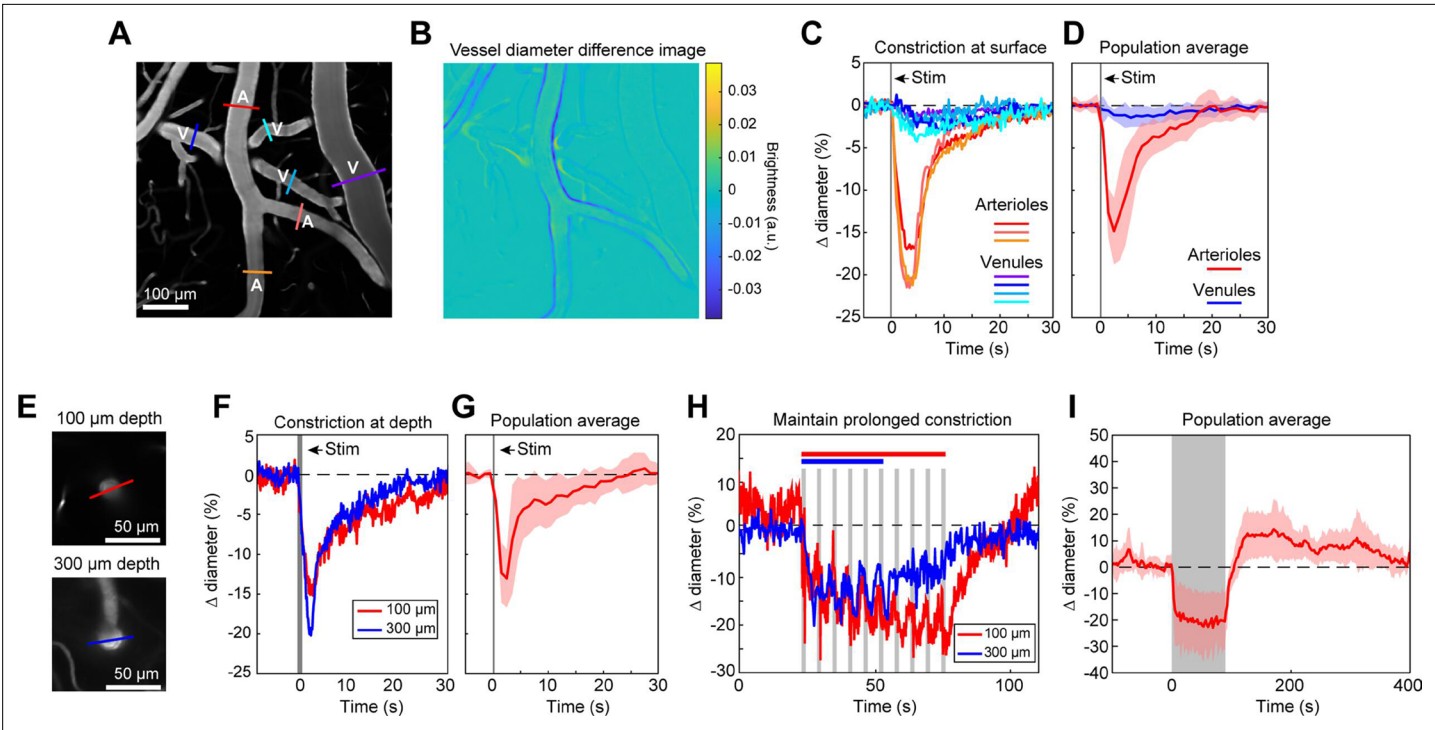

**Figure 2.** Full-field optogenetic activation of cortical vessels with 594 nm light-emitting diode (LED). (**A**) Cortical surface vessels labeled with fluorescein isothiocyanate (FITC) dextran. Arterioles (A) and venules (V) are marked. (**B**) Difference image created by subtracting the average of 4 s of data frames following stimulation (2.5–6.5 s after onset) from the average before stimulation (0–4 s before onset). Blue lines on vessel walls opposite each other indicate reduced brightness showing where constriction occurred. Vessels with yellow on one side and blue on the other indicate XY shifts in the image between the two intervals. Units are difference in arbitrary brightness values. (**C**) Time courses of vessel diameter changes. Colored lines correspond to cross-sections in panel A, with arterioles in varying shades of red and venules in shades of blue. Vertical gray band is optical stimulation interval (100 ms pulse). (**D**) Population average of 13 surface arterioles (red) and 10 surface venules (blue) from three mice stimulated with a single 100 ms pulse from the LED. Population constriction from 1 to 5 s following stimulation pulse: Arterioles = 11.1 ± 1%; Venules = 0.7 ± 0.2% (mean ± standard error of the mean [SEM]). (**E**) Images of a penetrating arteriole 100 µm deep (top) and 300 µm deep (bottom) in the cortex. Colored cross-sections correspond to time course traces in F and H. (**F**) A single train of LED light pulses (5 pulses, 100 ms, with 100 ms between) evoked strong constriction at both depths. (**G**) Population average constriction of five penetrating arterioles from three mice to a single 100 ms pulse (8.6 ± 1.7%). (**H**) We maintained vasoconstriction with repeated stimulation trains (5 pulses, 100 ms, 100 ms interpulse interval, 4 s between trains). Six trains were applied to the vessel at 300 µm and 10 to the vessel at 100 µm. (**I**) Population average of 11 vessels (pial and penetrating arterioles were combined as responses were similar) from three mice to 90 s of continual LED light pulses (100 ms pulses, 0.4 Hz).

of stimulation and flow completely stopped for approximately 2 s before returning to normal by 6.5 s poststimulation (*Figure 3B*). The penetrating arteriole rapidly constricted and the flow also stopped briefly around 2 s poststimulation (*Figure 3B*). Altogether, these results show that single-photon activation of ReaChR in smooth muscle cells causes rapid and robust vasoconstriction and reduced blood flow to the parenchyma.

## Two-photon activation of ReaChR-expressing mural cells produces focal vasoconstrictions

Two-photon light provides greater spatial resolution and targeting of deeper vessels compared to the single-photon activation achieved with the LED. To investigate the spatial control achievable with two-photon optogenetic activation of arterioles, we used a spatial light modulator (SLM) (*Chen et al., 2018*; *Yang and Yuste, 2018*) to split a single 1040 nm stimulation beam into multiple beamlets that could be focused on specific locations of interest. With this method, vasoconstriction was evoked only when the stimulation spots were contacting the vessel wall (*Figure 4A–E*). No vasoconstriction was evoked when the edges of the stimulation spots were 10 µm from the vessel wall, indicating high spatial precision in the X–Y plane. Stimulating pial arterioles led to robust constrictions that were constrained to a few tens of microns around the stimulation spots. We were able to constrict two

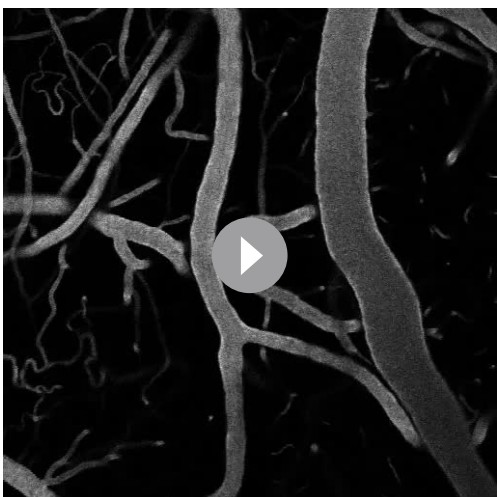

**Video 1.** Full-field activation with the 594 nm light-emitting diode (LED) in a PDGFRβ/ReaChR mouse. Image is looking down at pial surface vessels in the neocortex labeled with fluorescein isothiocyanate (FITC) dextran. A single 100 ms pulse from the 549 nm LED was used to stimulate the vessels. In this and other videos, the dark flash indicates the time of the optical stimulation. The image darkens because the detectors are briefly blocked by mechanical shutters to protect them from the intense stimulation light. This and subsequent videos are presented at 4× real time. Video is 755 µm x 755 µm window.

https://elifesciences.org/articles/72802/figures#video1

seen earlier with single-photon activation (*Figure 2B–D*) was likely a passive effect of the widespread arteriolar constriction.

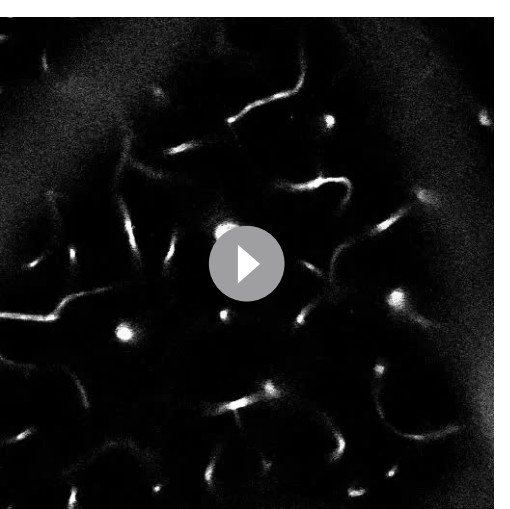

**Video 2.** Constriction of a penetrating arteriole 100 µm below the surface with the 594 nm light-emitting diode (LED). 5 pulses, 100 ms duration with 100 ms in between were presented. Video is a 378 µm x 378 µm window.

https://elifesciences.org/articles/72802/figures#video2

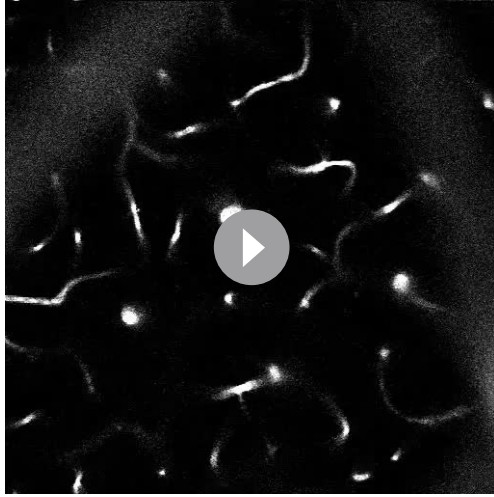

**Video 3.** The same vessel as *Video 2* was stimulated with 10 sets of 5 pulses, with 4 s between the onset of the pulse trains. Power and duration of pulses as in *Video 2*.

https://elifesciences.org/articles/72802/figures#video3

neighboring branches of a pial arteriole, separated by ~100 µm, independently with no overlap of constrictive responses (*Figure 4F–I*; *Videos 4 and 5*). We also used this spatial precision to directly target venules independently of arterioles. Venule branches did not constrict to two-photon activation (*Figure 4—figure supplement 1*), supporting our assertion that the constriction

Two-photon stimulation also allows targeting of individual vessels deeper in the cortex. We imaged a region with two penetrating arterioles 200 µm below the cortical surface. By splitting the stimulation beam and positioning spots over each vessel, we were able to constrict each vessel independently of the other (*Figure 4J–M*), showing precise control of vascular diameter between neighboring penetrating arterioles.

## Minimal spread of constriction along pial arterioles regardless of laser power

We next assessed the spatial precision of constriction along the length of the vessel. The difference images in *Figure 4B, C and G, H* demonstrated that there was very limited spread of the constriction beyond the stimulation spots. We tested the effect of laser power on the spatial spread of constriction by stimulating with a broad range of power levels. We found that increasing the laser power led to a small increase in the spread of constriction. For example, a 30-fold increase in

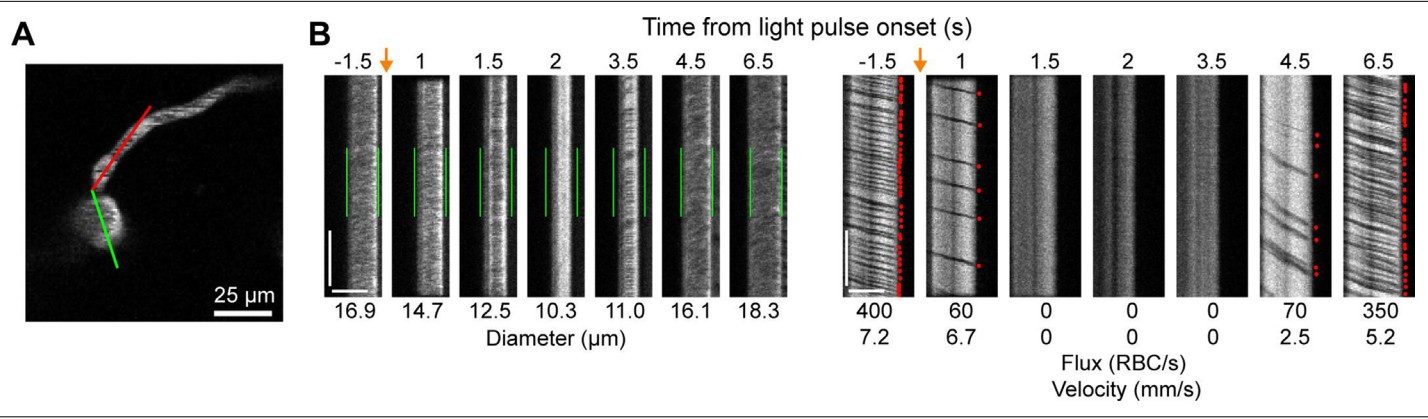

**Figure 3.** Optogenetic activation of vascular mural cells reduces blood flow. (**A**) Image of penetrating arteriole with an arteriole–capillary transition branch 85 μm below the surface. Line scans were acquired to measure the diameter of the arteriole (green segment) and the flow of the transition branch (red segment). A train of pulses (10 pulses, 100 ms, with 100 ms between) was delivered over the cranial window from the light-emitting diode (LED) (594 nm). (**B**) Line-scan data from seven time points during the run. *Y*-Axis is time with each row being a single scan of the laser line. *X*-Axis is distance along the line. The left set of panels shows the diameter data (the green segment in A) with the green bars indicating the prestimulus diameter. The right panels show the flow (the red segment in A) with each dot representing a single RBC passing through the segment (the dark streaks). The orange arrows indicate the interval when the light stimulus was applied. Numbers below each panel give the diameter of the penetrating arteriole and the flux and blood velocity for the transitional vessel. The arteriole constricts and flow is briefly eliminated (absence of dark spots in diameter segment at 2 s) before returning as the arteriole dilates back to the baseline level. Flow is reduced in the transition branch and then drops to zero (absence of dark streaks) before slowly recovering. Scale bars: time axis: 25 ms; distance axis: 25 μm.

power (from 5 to 150 mW total power) led to ~threefold increase in the spread of constriction (from ~25 to ~75 μm) (*Figure 5A–H*). Increasing the power to ~230 mW at this site led to damage of the vessel wall. Although the threshold for damage varied depending on vessel size and the number and arrangement of the stimulation spots, we frequently observed that high powers (typically >200 mW total across four to six spots) on pial arterioles caused i.v. dye extravasation, and this was occasionally accompanied by widespread vasoconstriction (*Figure 5—video 1*). Laser-induced vascular injury was also sometimes observed at very high powers (>250 mW) in control mice expressing only YFP in mural cells (see *Figure 5—videos 2; 3*). However, at laser powers routinely used to induce vasoconstriction in ReaChR mice (<200 mW), we never observed constrictions in control mice (*Figure 5—figure supplement 1*; *Figure 5—videos 4; 5*). This confirms that the focal constrictions seen at lower laser powers are ReaChR dependent and not attributable to direct effects of light on blood vessels (*Rungta et al., 2017*; *Rorsman et al., 2018*).

## Stimulation power influences the axial resolution of photoactivation

The precision of two-photon photoactivation in 3D also depends on the resolution in the axial dimension, where the point-spread-function is larger than in the lateral dimension (*Rickgauer and Tank, 2009*; *Packer et al., 2015*). To measure the axial resolution, we used the SLM to position stimulation spots at multiple depth planes above and below a pial arteriole while imaging the center of the arteriole (*Figure 6A, B*). When spots were focused 100 μm above or below the vessel at 115 mW total power, the vessel constricted approximately 25–35% of the magnitude caused by direct stimulation of the vessel. At 150 μm below the vessel, constriction dropped to ~10% (*Figure 6C, D*). Constriction was greater when the stimulation was above the pial surface than when it was deeper in the tissue, likely due to increased light scattering when penetrating the cortical tissue (*Figure 6D* – compare 'above pia' with 'parenchyma').

To ensure that the SLM was accurately focusing the laser power throughout the depth range, we bleached a fluorescent slide with SLM stimulation at the imaging plane and 200 μm above and below the imaging plane. This confirmed that the region of excitation was accurately shaped and placed regardless of the SLM focal plane (*Figure 6—figure supplement 1*).

Next, we tested the effect of laser power on the axial range of constriction. While the higher laser power used in *Figure 6C, D* showed substantial axial spread, lower laser powers (20–44 mW) resulted

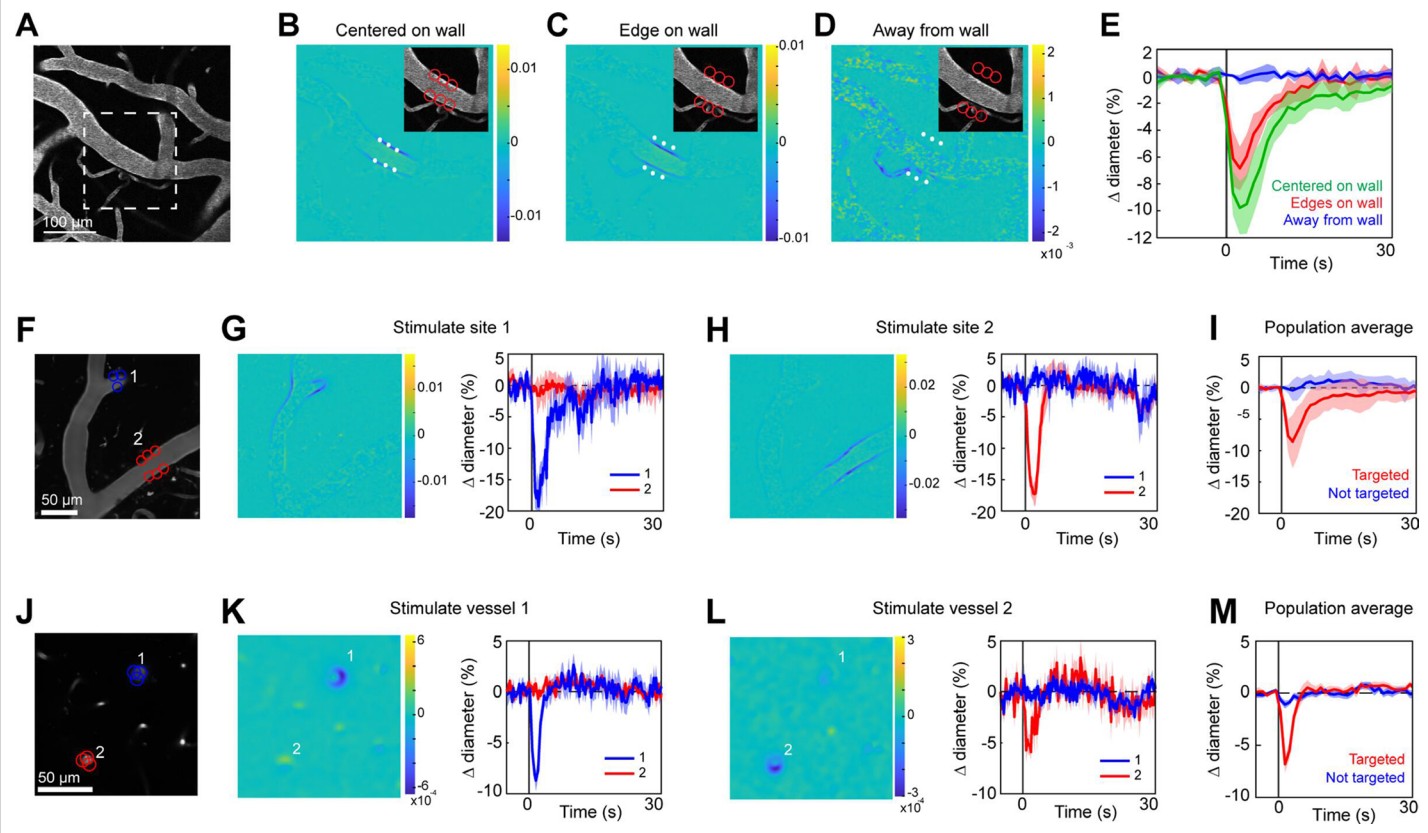

**Figure 4.** Spatial precision of two-photon optogenetic activation in the lateral plane. (**A–E**) *XY*-precision of photostimulation. (**A**) Image of surface vessels labeled with Texas Red-dextran in an opsin mouse that were photostimulated with spots in different positions relative to the vessel wall. Dashed rectangle indicates region in inset on the difference image panels (**B–D**). Three spot positions were used: (**B**) spots centered on vessel walls, (**C**) spot edges on vessel walls, and (**D**) spot centers one diameter away from vessel walls. Difference images as in *Figure 2*. Red circles indicate the location and size of the spots, white asterisks indicate spot centers. (**E**) Time courses of diameter changes based on spot position. Average of three vessels, five repetitions each. Error bands are standard deviation (SD). Total laser power was ~130 mW divided between the six spots. Mean constriction for the three conditions was 7.6%, 4.8%, and 0.0% for B, C, and D respectively, with standard error of the mean (SEM) <1% for all three conditions. (**F–I**) Stimulation of different surface branches leads to isolated constriction. (**F**) Surface vasculature labeled with Texas Red-dextran. Colored circles indicate size/location of stimulation spots and correspond to time course plots in G, H. For G, total laser power was ~90 mW. For H, total power was ~70 mW. (**G, H**) Left panels: Difference images showing constriction (blue lines on both sides of vessel) only near stimulation points. Right panels: Time courses of diameter change for the two different stimulations: Error bands are SD across five (**G**) or four (**H**) repetitions. Vertical gray band is stimulation interval (100 ms). (**I**) Population average constriction of vessel segments when targeted (red) or not (blue). *N* = 8 vessels in 3 animals. Average total power per vessel was 138 mW (range 90–250 mW). Population mean ± SEM from 1 to 5 s following stimulation: Targeted arterioles = 6.5 ± 1%; Untargeted = 0.3 ± 0.4%. (**J–M**) Stimulation of penetrating arterioles with single-vessel precision. (**J**) Two penetrating arterioles 200 µm below the cortical surface. Left panels: Difference images when stimulating arteriole #1 (**K**) or arteriole #2 (**L**). Each arteriole was stimulated with a total power ~110 mW. Right panels: Time courses of diameter change of the vessels indicated in the left panels. Data presented as mean and SD across seven repetitions. (**M**) Population average constriction of penetrating arterioles when targeted with stimulation (red) and when not targeted (blue). *N* = 9 vessels in 3 animals. Average total power applied to each vessel was 130 mW (range 80–190 mW). Population mean ± SEM from 1.5 to 4.5 s following stimulation. Targeted arterioles = 3.8 ± 0.6%; Untargeted = 0.7 ± 0.3%.

The online version of this article includes the following figure supplement(s) for figure 4:

**Figure supplement 1.** Two-photon optogenetic activation in venules versus arterioles.

in pial arteriole constriction which was almost exclusively restricted to the stimulation plane, although the magnitude of the constrictive response was reduced (*Figure 6E*).

An important consideration in determining out-of-focal-plane activation, is that the cone of light extends laterally outside of the focal plane. This could lead to a greater lateral spread of activation outside the focal plane than seen within the plane in *Figure 4E*. Therefore, we repeated the lateral resolution test with high laser power (115 mW) at different depth planes. When the stimulation spots were laterally offset from the vessel wall, vasoconstriction was greater when stimulating

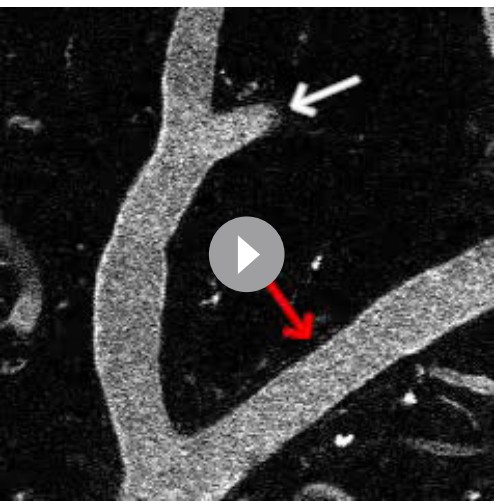

**Video 4.** Two-photon activation leads to focal constriction. The spatial light modulator (SLM) focused six spots at the location indicated by the red arrow. The total stimulation power was ~200 mW spread over the six spots with three adjacent spots on either side of the vessel. Each spot was scanned in a 12-μm diameter spiral for 100 ms. The white arrow is for comparison with *Video 5*. Vessels are labeled with Texas Red-dextran. Video is a 240 μm x 240 μm window.

https://elifesciences.org/articles/72802/figures#video4

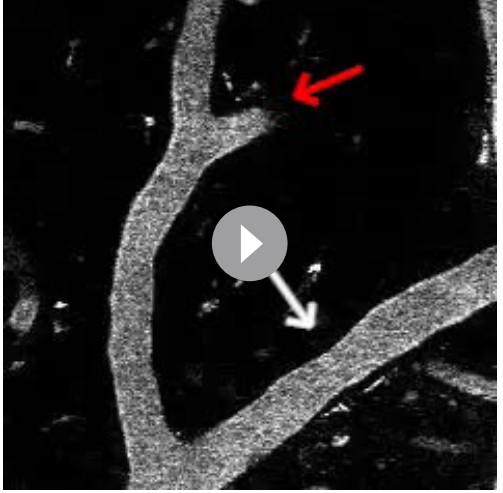

**Video 5.** The same field-of-view as *Video 4* but here a different branch of the pial artery is stimulated (red arrow). Total power was ~75 mW spread out over three spots positioned on the branch stub and three off-target spots. The white arrow indicates the location stimulated in *Video 4* for comparison.

https://elifesciences.org/articles/72802/figures#video5

50–150 μm below the imaging plane than in the imaging plane. However, just a 10-μm lateral offset resulted in very small vasoconstriction at all depths except when very high power levels were used (200 mW) (*Figure 6F*).

Thus, for precise 3D targeting of vasoconstriction, care must be taken to avoid unwanted activation of vessels within the light cone but outside the focal plane. The range of out-of-focus activation will depend not only on the laser power applied, but also on the objective used, the orientation of the vessel relative to the light cone, and the size of the arteriole.

## Vasoconstriction in the axial plane along penetrating arterioles

We next measured the spread of constriction in the axial plane through penetrating arterioles. We stimulated penetrating arterioles 200 μm below the surface with a range of powers and imaged in the stimulation plane and 150–175 μm superficial to that plane (*Figure 7A–C*). We found that lower powers (20–30 mW) could constrict penetrating arterioles in the stimulation plane with little spread to the superficial plane. With moderate powers (~70 mW), near maximal constriction could be achieved but this led to ~50% constriction in the superficial plane. With higher power (>150 mW), near maximal constriction could be achieved in both planes. Since penetrating arterioles supply columns of cortical microvasculature (*Shih et al., 2013*), the use of high powers to constrict a large span of the penetrating arteriole could be useful for modulating blood flow to a column of cortical tissue in a reversible manner (*Figure 7—figure supplement 1*).

## Optogenetic vasoconstriction is slower and smaller in microvessels

Previous work has shown that prolonged optogenetic stimulation of pericytes expressing ChR2 leads to slow constriction of capillaries (*Nelson et al., 2020*; *Hartmann et al., 2021*). We therefore tested the efficacy of ReaChR for optogenetic constriction of pericytes. We divided vessels into three categories: penetrating arterioles (zero order; covered by α-smooth muscle actin (SMA)-positive smooth muscle cells), ACT vessels (branch orders 1–4; covered by α-SMA-positive ensheathing pericytes), and capillaries (branch orders 5–9; covered by α-SMA-low/undetectable capillary pericytes)(*Figure 8A*; *Hartmann et al., 2021*). Brief pulses of single-photon excitation with the 594 nm LED led to ~14% constriction of penetrating arterioles, ~5% constriction of ACT vessels, and a small ~1% constriction of capillaries (*Figure 8B*). Single pulses of two-photon stimulation, which elicited

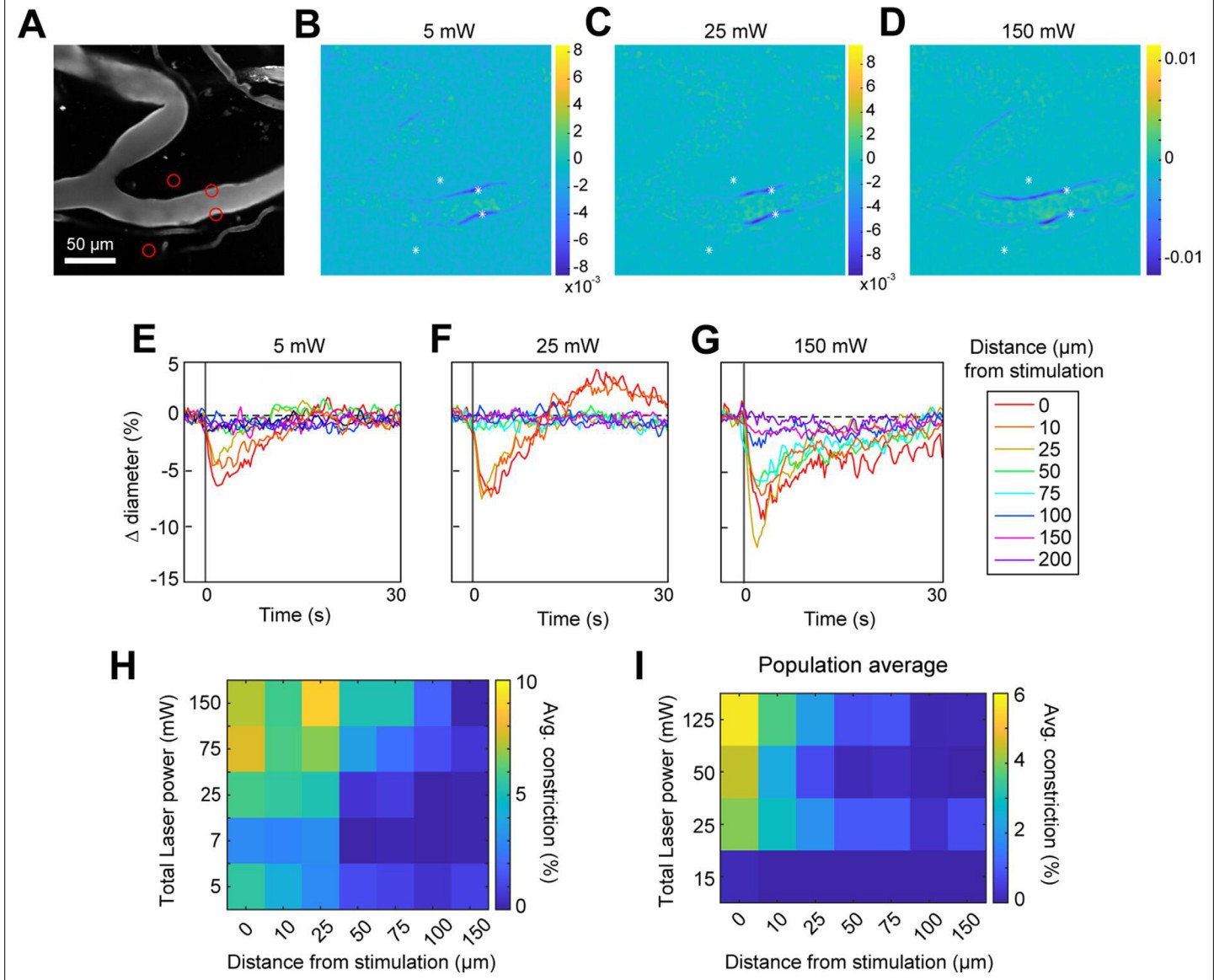

**Figure 5.** Effect of laser power on spread of constriction. (**A**) Vessels labeled with Texas Red-dextran. Red circles show spot size and location. Two spots off the vessel were to maintain similar power levels per spot compared to other runs (not shown). (**B–D**) Difference images for three different power levels (values indicate total power of all four spots). (**E–G**) Time courses of diameter change at the three power levels in B–D at different distances along the arteriole from the center of stimulation. (**H**) Summary of constriction at all power levels tested across different distances. Average constriction from 1 to 5 s following stimulation onset (higher numbers/yellow color indicates greater constriction). (**I**) Population average from three vessels in three mice of constriction at different power levels and distances from stimulation location. Six spots were placed on vessels (15 μm, three on each side of vessel).

The online version of this article includes the following video and figure supplement(s) for figure 5:

**Figure supplement 1.** Control mouse expressing YFP only in mural cells exhibit no vasoconstriction.

**Figure 5—video 1.** Excessive laser power leads to vessel damage and widespread constriction.
https://elifesciences.org/articles/72802/figures#fig5video1

**Figure 5—video 2.** Control mouse expressing YFP in vessel walls shows that excessive laser power can lead to widespread constriction and vessel damage independently of opsins.
https://elifesciences.org/articles/72802/figures#fig5video2

**Figure 5—video 3.** Another example of vessel trauma at high powers in control mouse.
https://elifesciences.org/articles/72802/figures#fig5video3

**Figure 5—video 4.** Control mice show no constriction after light pulse when using powers that resulted in clear constrictions in ReaChR mice.

*Figure 5 continued on next page*

*Figure 5 continued*

https://elifesciences.org/articles/72802/figures#fig5video4

**Figure 5—video 5.** Same as Figure 5 - Video 3 but the power was ~200 mW.

https://elifesciences.org/articles/72802/figures#fig5video5

robust constriction in penetrating arterioles, led to minimal (~1%) constriction of ACT vessels and no constriction in capillaries (*Figure 8C*). To determine if more prolonged stimulation of ReaChR would induce constriction, we applied repeated pulses of light with the SLM for 40 s (*Figure 8D*). While ACT vessels constricted by ~3–4%, capillaries still showed negligible constriction. We then considered the duty cycle of stimulation. Our prior studies stimulated pericytes using line scanning for near constant laser stimulation (*Hartmann et al., 2021*), whereas the SLM pulses used here were 100 ms with 0.5–1.5 s between pulses. Continuous line-scan stimulation at 1045 nm led to rapid constriction in penetrating arterioles and slower constrictions in the other vessel categories (*Figure 8E*). Consistent with our prior findings (*Hartmann et al., 2021*), capillaries exhibited a gradual constriction over 60 s of stimulation. ACT vessels exhibited a more rapid constriction in the early phase. Constriction of both capillaries and ACT vessels reached a maximum of ~10% (*Figure 8E*). These results show that near continuous stimulation provided by line scans is necessary to constrict the smallest microvessels.

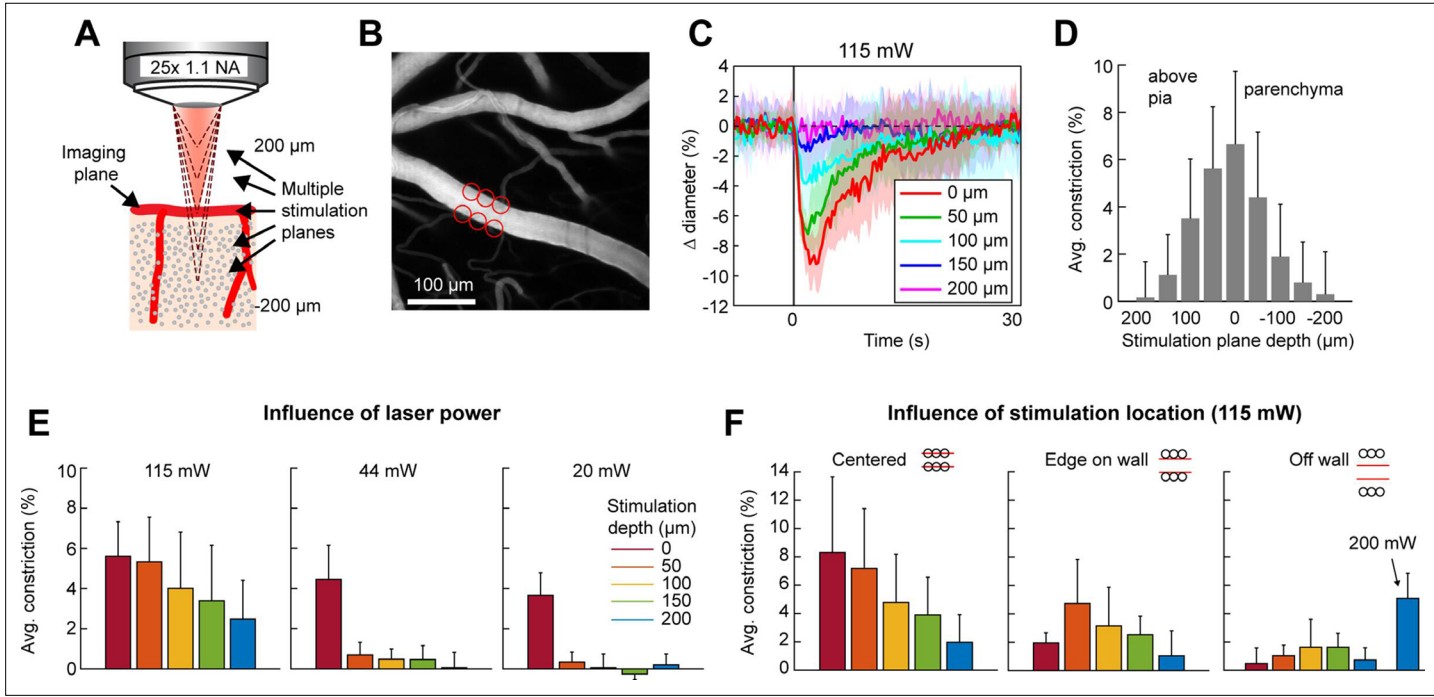

**Figure 6.** Axial resolution of two-photon optogenetic activation. (**A**) Schematic of imaging setup for measuring the axial resolution of two-photon activation. The objective focused the imaging beam on a pial arteriole at the cortical surface. The dashed lines indicate various focal planes of the stimulation laser generated by the spatial light modulator (SLM). In the experiment, stimulation spots were focused from 200 μm above to 200 μm below the imaging plane in 50 μm steps. (**B**) Projection of surface vasculature labeled with Texas Red-dextran. Red circles indicate size and *XY* position of stimulation spots in varying depth planes. Total power was ~115 mW. (**C**) Time courses of diameter changes following stimulation at different depths from the surface. Mean and standard deviation (SD) across eight repetitions. Equidistant planes above and below the surface were averaged together for visibility. (**D**) Constriction amount at the different depths computed as the average change in diameter across repetitions from 1 to 5 s following the stimulation pulse. Error bars are SD across eight repetitions. (**E**) Axial resolution at three different total power levels averaged across three surface arterioles from three mice. Error bands are SD across vessels. (**F**) Influence of stimulation spot location at different depths. 115 mW total power. *N* = 4 surface arterioles from 4 mice. For the 'Off wall' condition, 200 mW total power was also used at the 200 μm depth.

The online version of this article includes the following figure supplement(s) for figure 6:

**Figure supplement 1.** Spatial light modulator (SLM) focus test.

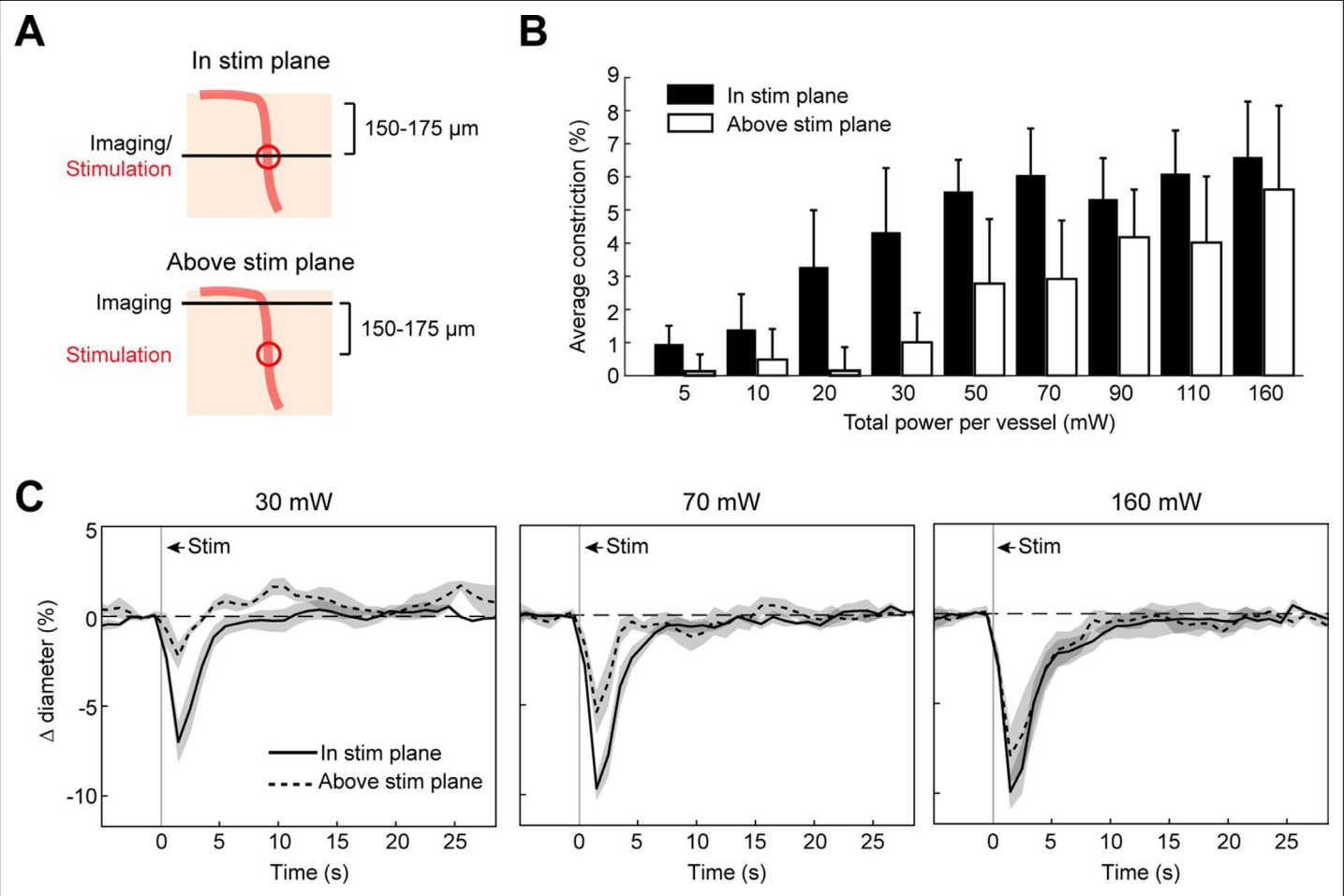

**Figure 7.** Axial spread of two-photon optogenetic activation. (**A**) Penetrating arterioles were stimulated using the spatial light modulator (SLM) (100 ms light pulse) 200 µm below the surface while imaging was performed at the stimulation plane (top) or 150–175 µm above the stimulation plane (bottom). (**B**) Average constriction from 0.5 to 3.5 s after stimulation across a range of powers. Three 15 µm diameter spots were placed on each arteriole and the total power to vessel is reported. (**C**) Time course of constriction at the stimulation plane and above the stimulation plane for penetrating arterioles at three different stimulation power levels. Data for B and C are average and standard deviation (seven repetitions) of four arterioles from two mice.

The online version of this article includes the following figure supplement(s) for figure 7:

**Figure supplement 1.** Broad axial constriction of penetrating arterioles with higher stimulation power.

### Faster vasoconstriction with mural cell activation of ReaChR compared to ChR2

We next compared the contractile kinetics of mural cells expressing ReaChR with our previous data from mice expressing ChR2 in mural cells (*Hartmann et al., 2021*; *Figure 9A, B*). We computed the rate of constriction over the first 10 s of stimulation for capillaries and ACT vessels (*Figure 9C*). Since penetrating arterioles expressing ReaChR constricted rapidly and their responses had nearly plateaued by 10 s, we computed the rate of constriction for the first 5 s of stimulation for the penetrating arterioles (*Figure 9C*). Capillaries showed similar, slow contractile dynamics with both opsins, but ACT vessels and penetrating arterioles exhibited more rapid initial constriction with ReaChR compared to ChR2 vessels.

### Optogenetic vasoconstriction is attainable in awake animals

To ensure that vessels could also be precisely targeted in awake animals, we replicated some key findings in awake mice. As in the anesthetized animals, brief pulses of light from the LED evoked rapid robust constrictions and the constrictions could be maintained with repeated light pulses (*Figure 10A, B*). Additionally, two-photon activation provided single-vessel precision of vasoconstriction (*Figure 10C*).

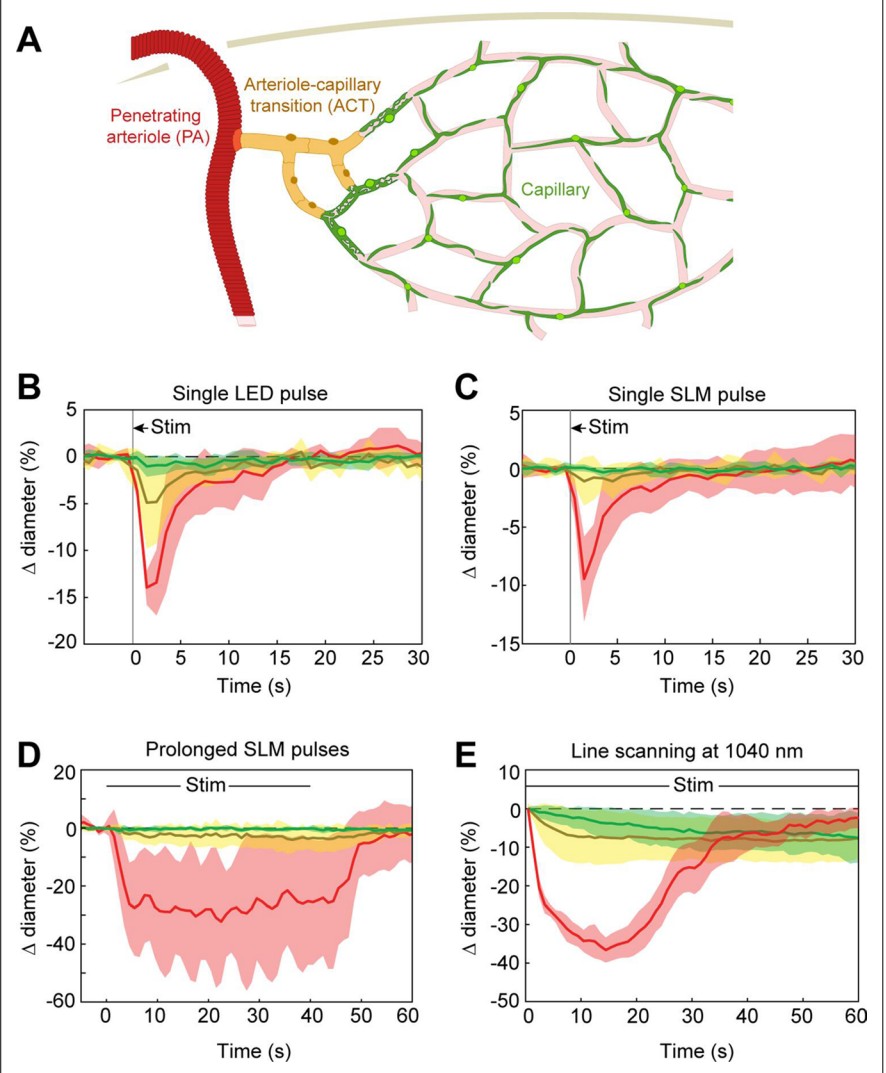

**Figure 8.** Optogenetic stimulation of vessels across microvascular zones. (**A**) Schematic showing three microvascular zones examined with different stimulation paradigms. (**B**) Time course of constriction during a single 100 ms light-emitting diode (LED) pulse ($N$ = 4 penetrating arterioles, $N$ = 6 arteriole–capillary transition (ACT) vessels, and $N$ = 9 capillaries). (**C**) Time course of constriction during a single 100 ms pulse using the spatial light modulator (SLM) (100–130 mW total power on each vessel) ($N$ = 7 penetrating arterioles, $N$ = 5 ACT vessels, and $N$ = 4 capillaries). (**D**) Time course of constriction to repeated SLM pulses (100 ms pulses, 0.8–1.8 Hz, 40 s) ($N$ = 3 penetrating arterioles, $N$ = 6 ACT vessels, and $N$ = 5 capillaries). (**E**) Time course of constriction to continuous line scanning ($N$ = 9 penetrating arterioles, $N$ = 24 ACT vessels, and $N$ = 15 capillaries). Image collection and photoexcitation were simultaneously achieved with 1040 nm line scanning across vessels. Laser power varied from 10 to 120 mW with higher powers used for deeper vessels.

In general, there was a slight decrease in constriction amplitude and faster return-to-baseline in awake versus anesthetized animals.

## Use of optogenetic stimulation to modulate sensory-evoked dilation

We next examined the utility of this technique to directly modulate the physiological dynamics of brain arterioles, thereby providing a novel means to dissociate vascular and neuronal responses in processes such as functional hyperemia and vasomotion. Modulation of physiological dynamics is possible because optogenetic activation of mural cells evokes rapid contraction within hundreds of milliseconds (*Figure 11—figure supplement 1*), compared to the slower dynamics of neurovascular coupling, which occurs on the order of seconds (*Silva et al., 2000*; *Hillman, 2014*). We directly

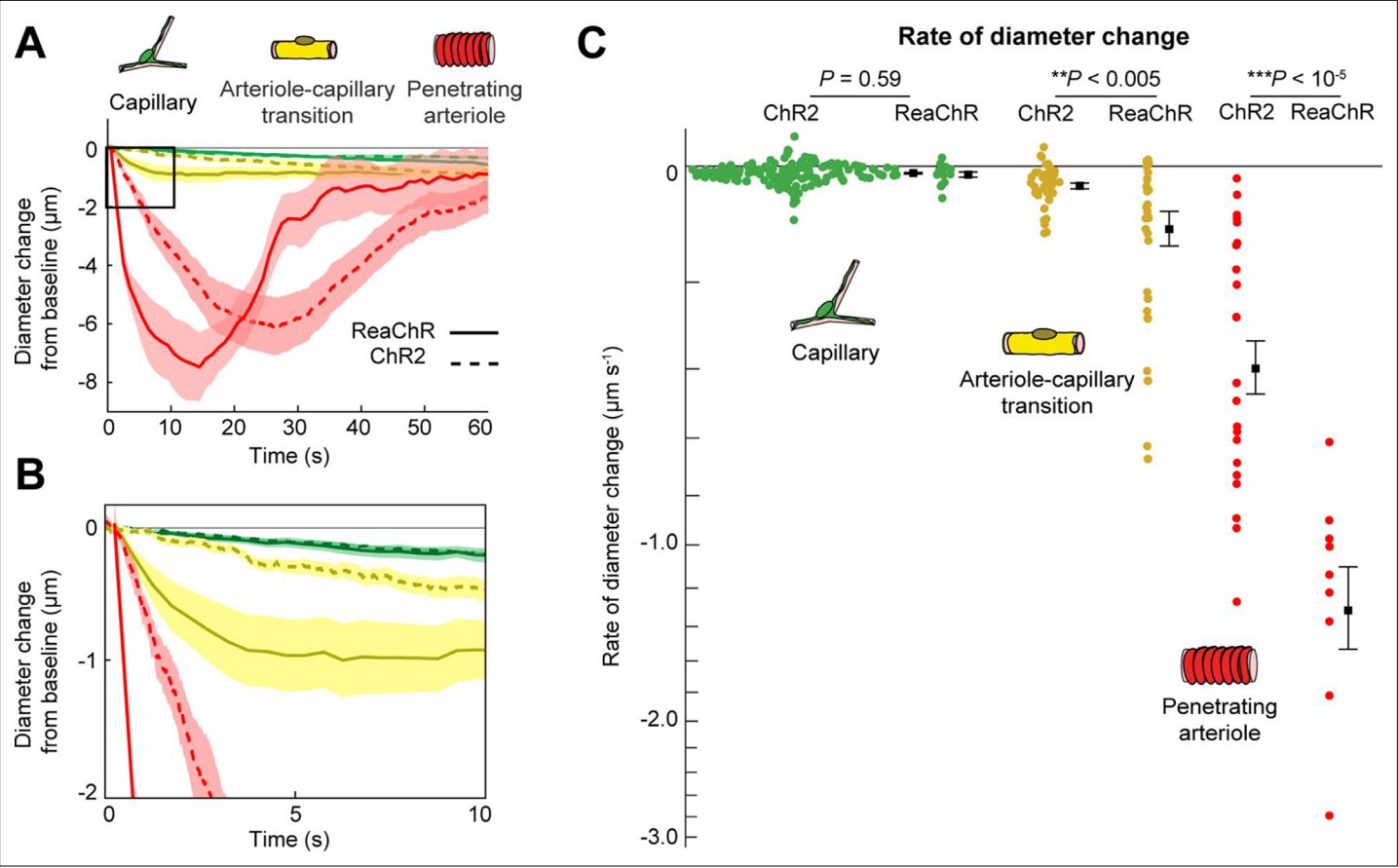

**Figure 9.** Comparison of ReaChR and ChR2 activation. (**A**) Time course of constriction of three vessel classes covered by different mural cell types (ChR2 data from *Hartmann et al., 2021*; ReaChR data from line-scan stimulation in *Figure 8E*). Error bands are standard error of the mean (SEM). Peak constriction time for penetrating arterioles: ReaChR = 14.5 s, ChR2 = 26 s. (**B**) Inset showing zoomed view of the first 10 s of data. (**C**) Rate of diameter change for each vessel computed over the first 10 s for capillaries and arteriole–capillary transition (ACT) vessels, and over the first 5 s for penetrating arterioles. Black squares and error bars are mean and SEM. Mean values by group: Capillaries (−0.012 to −0.010 µm s$^{-1}$ for ReaChR and ChR2, respectively); Transitional vessels (−0.10 to −0.03 µm s$^{-1}$); Penetrating arterioles (−1.31 to −0.40 µm s$^{-1}$). Results from one-way analysis of variance (ANOVA) on opsin type for each vessel class are shown.

compared the response latency of optogenetically evoked constriction to visually evoked dilation in the same penetrating arterioles, and found much faster responses to the optogenetic stimulation (0.6 s to reach 25% of peak constriction) than to the visual stimulus (3.4 s to 25% peak dilation; *Figure 11A* – magenta vs. blue).

As a proof-of-principle experiment, we determined whether optogenetically driven constriction could alter sensory-evoked vasodilation. We presented 5 s of visual stimulation together with a brief 100 ms pulse of optogenetic stimulation. This led to a rapid constriction of the vessel followed by a dilation that was lower in amplitude than the dilation to the sensory stimulus alone (*Figure 11A, B*, green vs. blue). By reducing the visual stimulus duration from 5 to 2 s, the same light pulse completely suppressed the dilation phase of the vascular response following the initial rapid constriction (*Figure 11A, B*, red vs. blue). Similarly, using multiple pulses of light during the visual stimulation was also sufficient to eliminate the sensory-evoked vasodilation, as shown in population data across animals (*Figure 11—figure supplement 2*). Thus, vasodilation to a sensory stimulus can be negated by optogenetic 'clamping' of arteriole diameter.

## Discussion

Building on techniques recently developed for all-optical interrogation of neurons (*Packer et al., 2015*; *Carrillo-Reid et al., 2016*; *Marshel et al., 2019*), we introduce a technique for all-optical

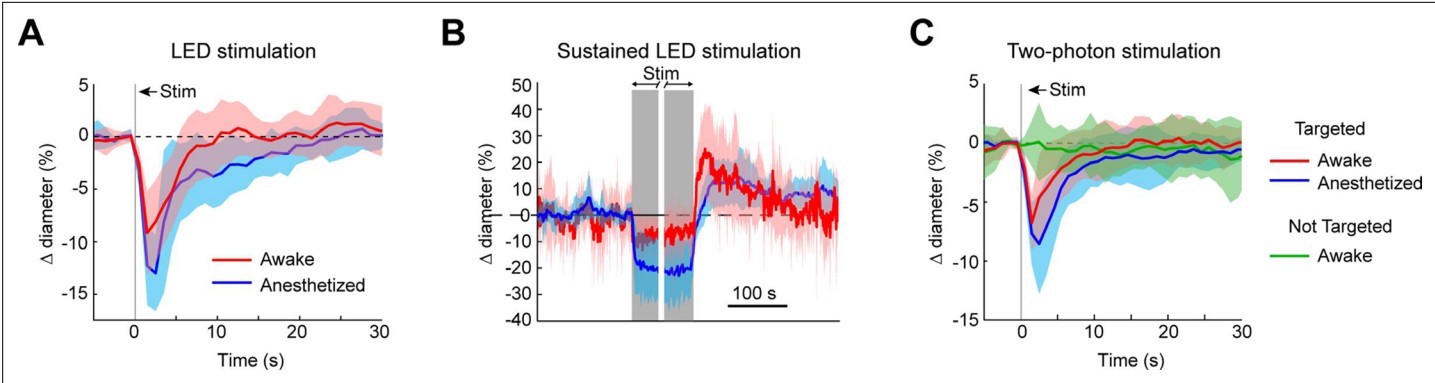

**Figure 10.** Optogenetic activation of vasculature in awake versus anesthetized mice. (**A**) Constriction of surface arterioles in awake and anesthetized mice to a 100 ms pulse from the light-emitting diode (LED). Anesthetized data from *Figure 2D*. Awake: *N* = 16 vessels in 2 animals. Population mean ± standard error of the mean (SEM) from 1 to 5 s following stimulation: 6.5 ± 0.6%. (**B**) Constriction to prolonged repeated pulses (100 ms, 0.43–0.6 Hz, 56–145 s). Since different stimulus durations were used across animals, the time courses are aligned by onset and then by offset. *N* = 14 vessels from 7 animals. Anesthetized data from *Figure 2I*. (**C**) Single-vessel precision in awake animals. Average constriction of 14 vessels from 3 animals to a 100 ms pulse of two-photon stimulation using the spatial light modulator (SLM) when targeted with the light (red) and when not targeted (green). Population mean ± SEM: Targeted arterioles = 4.3 ± 0.5%; Not targeted = 0.3 ± 0.5%. Anesthetized data from *Figure 4I*.

interrogation of the brain vasculature. We expressed the red-shifted opsin ReaChR in vascular mural cells to control arteriole diameter rapidly and reversibly over a range of spatiotemporal scales in vivo. Single-photon activation produced widespread vasoconstriction across the full cranial window, whereas two-photon activation provided single-vessel control of constriction in 3D. Single brief pulses of light could produce robust vasoconstriction with rapid onset, and repeated light pulses could maintain prolonged constriction. Optical stimulation of mural cells expressing ReaChR is therefore a powerful approach for studying arteriole vasoconstriction and the spatiotemporal modulation of cortical blood flow, and for dissociating neuronal and vascular activity in complex processes such as neurovascular coupling and vasomotion.

Prior studies have used the original variant of channelrhodopsin, ChR2 (H134R), to achieve vasoconstriction in vascular mural cells in vivo. One- and two-photon activation of ChR2 led to the gradual constriction of arterioles, requiring >20 s to reach a minimum diameter with near-constant activation

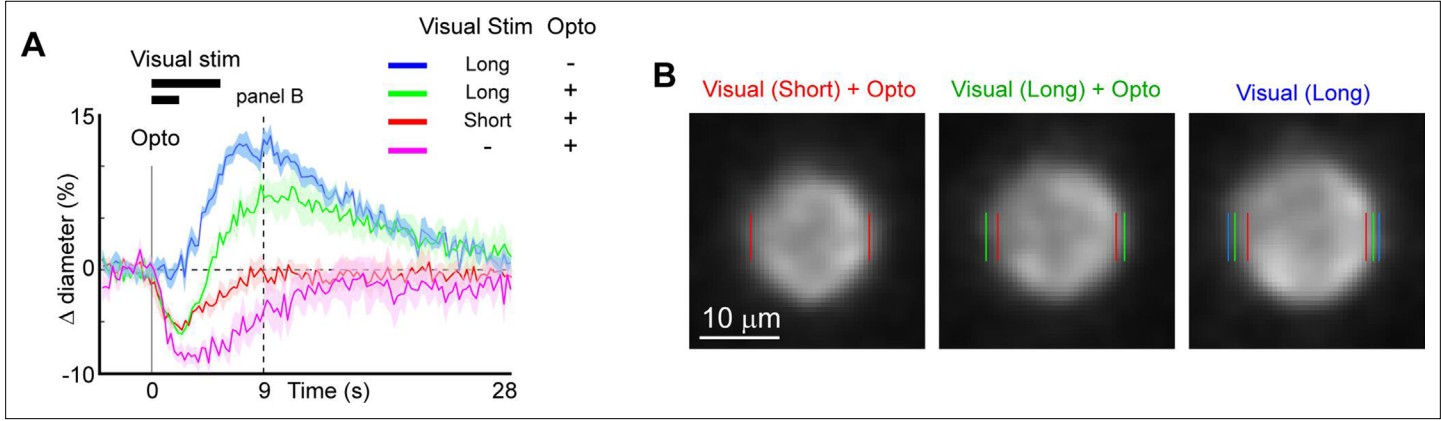

**Figure 11.** Modulation of sensory-evoked vasodilation using optogenetics. (**A**) Pairing optogenetic stimulation with visual stimulation leads to reduced (green) or eliminated (red) visually evoked dilations compared to visual stimulation alone (blue). Vertical gray band at time 0 is optogenetic stimulation interval (100 ms) and black horizontal bars indicate visual stimulation duration (5 or 2 s, corresponding to long and short, respectively). Vertical dashed line at 9 s indicates time point of images shown in B. (**B**) Single-frame images showing penetrating arteriole 9 s after stimulus onset for the conditions with visual stimulation. Colored lines indicate the computed diameter of the vessel cross-section in the image.

The online version of this article includes the following figure supplement(s) for figure 11:

**Figure supplement 1.** Speed of optogenetic constriction.

**Figure supplement 2.** Offsetting functional hyperemia with optogenetic activation of mural cells – population data.

(*Hill et al., 2015*; *Tong et al., 2020a*; *Hartmann et al., 2021*). In contrast, continuous activation of ReaChR in penetrating arterioles (using laser line scanning) led to a more rapid constriction, reaching a minimum in <15 s, that was greater in amplitude and returned to baseline more quickly. This suggests that ReaChR is superior to ChR2 for two-photon optogenetic studies of the vasculature, likely because of the higher photocurrent amplitude in ReaChR, compared to ChR2 (H134R), at their respective optimal excitation wavelengths (*Lin et al., 2013*). The improved tissue penetration of 1040 nm light used to activate ReaChR may also increase stimulation efficiency for parenchymal vessels (*Chaigneau et al., 2016*).

ReaChR is among the most potent red-shifted opsins for two-photon activation as shown in neurons (*Chen et al., 2019*), and its availability as a floxed mouse line allows vascular targeting experiments. Mural cells are not easily transduced in vivo using viral approaches. An additional advantage is that the optimal activation wavelength of ReaChR (>1000 nm) does not substantially interfere with imaging of common fluorophores, whereas two-photon ChR2 activation occurs around the imaging wavelengths used for calcium indicators such as Oregon Green or GCaMP (*Prakash et al., 2012*). This feature facilitates independent imaging of neural activity and vascular stimulation with separate laser lines.

The highly localized constriction seen following a focal activation differs from vasodilation responses seen during functional hyperemia. Following focal activation of neural tissue, vasodilatory signals propagate several millimeters from the activation site (*Iadecola et al., 1997*; *Chen et al., 2011*). This long-range propagation of dilation is attributed to endothelial cell signaling through gap junctions (*Chen et al., 2014*; *Longden et al., 2017*; *Zechariah et al., 2020*), and influences the spatial precision of vascular imaging (e.g., fMRI) signals (*O'Herron et al., 2016*; *Rungta et al., 2018*; *Drew, 2019*). The lack of propagation seen when directly activating smooth muscle cells supports the notion that endothelial rather than smooth muscle cells mediate the long-range propagation of signals through vessel walls. However, differences in the propagation characteristics of hyperpolarizing (dilatory) versus depolarizing (constrictive) signals along the vascular wall may also play a role in the propagation range of the signals.

We anticipate that this technique will be useful for studies requiring assessment of vascular dynamics in vivo. The ability to directly modulate vessel diameter and blood flow levels with high spatiotemporal precision could be useful for understanding vascular wall kinetics. We have shown that it is possible to study the contractile kinetics of different vessel types within the microvascular network. Our data confirm the distinct kinetics across arterioles, ACT, capillary and venous zones, with slow contraction occurring in capillaries covered by αSMA-low/undetectable pericytes. These slow kinetics have implications for how contraction of capillary pericytes contribute to blood flow regulation, which is relevant to maintenance of basal capillary flow and flow heterogeneity (*Hartmann et al., 2021*; *Berthiaume et al., 2022*; *Hartmann et al., 2022*).

The ability to concurrently monitor neural and/or astrocytic activity alongside vessels during vascular manipulation opens additional opportunities to study questions on neuro–glia–vascular coupling. For example, offsetting stimulus-evoked dilations (*Figure 11*) while monitoring neural activity can shed light on how neuronal function depends on functional hyperemia (*Moore and Cao, 2008*). Neural circuits are extremely complex with many different subtypes of neurons playing different roles. These subtypes have been shown to have different metabolic sensitivities and thus, may be differentially affected by blocking functional hyperemia (*Kann, 2016*). Additionally, the energy budgets of different cellular functions within neurons are quite different (*Howarth et al., 2012*) and reducing available energy by blocking functional hyperemia could lead to differing degrees of dysfunction across key cellular processes (e.g., re-establishing the membrane potential, recycling neurotransmitters). These changes could lead to altered circuit activity which could have profound consequences for neural processing. Furthermore, it has been shown that there is a steep gradient of oxygen moving away from penetrating arterioles, and so neurons at greater distances from vessels may be differentially affected by blocking the hyperemic response (*Devor et al., 2011*).

Optical control of arteriole tone can also be used to study 'vasculo-neuronal coupling', a process where alteration in vessel tone modifies neural and astrocyte signals. Theoretical (*Moore and Cao, 2008*) and brain slice (*Kim et al., 2016*) studies have proposed the existence of such processes. The direct, precise control of vessel diameter afforded by optogenetics provides a tool to study neuro–glia–vascular interactions in vivo. In addition, the role of vasomotion in clearance of metabolic waste

through perivascular pathways could also be studied in more detail with spatiotemporal modulation of arteriole diameter in vivo (*van Veluw et al., 2020*).

Impaired functional hyperemia is seen in numerous neurologic diseases, including acute injuries caused by ischemia (*Summers et al., 2017*), and during progressive, age-related pathologies such as cerebral amyloid angiopathy and other small vessel diseases (*Niwa et al., 2002*; *Park et al., 2014*). It is currently difficult to pinpoint the basis of impaired neurovascular responses, as defects could arise from loss of neuronal activation, vascular reactivity, coupling between these two processes, or a combination thereof. The ability to selectively block the vascular response with optogenetics provides cleaner access for probing neurovascular physiology and pathophysiology in vivo, independent of neural activity.

## Materials and methods

Experiments were performed at the Medical University of South Carolina (MUSC) and Augusta University (AU). All surgical and experimental procedures were approved by the Institutional Animal Care and Use Committees of the Universities (current AU protocol #0982).

### Animals

Mice were generated at our Institutions by crossing heterozygous PDGFRβ-Cre male mice (courtesy of Dr. Volkhard Lindner, Maine Medical Center Research Institute) with heterozygous female opsin mice (floxed ReaChR-mCitrine; Jackson Laboratories: Strain #026294). Control mice were generated by crossing the same PDGFRβ-Cre males with Ai3 females (floxed YFP; Jackson Laboratories: Strain #007903). Bigenic progeny from these crosses displayed green fluorescence in the vasculature under blue illumination in tail snips or ear punches. Male and female mice were used aged 2–12 months.

### Surgeries

Mice were anesthetized with a bolus infusion of fentanyl citrate (0.04–0.05 mg kg$^{-1}$), midazolam (4–5 mg kg$^{-1}$), and dexmedetomidine (0.20–0.25 mg kg$^{-1}$) several hours after an intramuscular injection (0.03 ml) of dexamethasone sodium phosphate (4 mg ml$^{-1}$). The heart and respiration rates of the animals were continually monitored throughout the surgeries using pulse oximetry (PhysioSuite, Kent Scientific). The scalp was excised, the skull cleaned, and a custom-made head-plate was fixed to the skull with C&B MetaBond quick adhesive cement (Parkell; S380). Craniotomies (2–3 mm) were opened over the primary visual cortex centered approximately 2.5 mm lateral to the lambda suture and 1–1.5 mm anterior to the transverse sinus. Craniotomies were sealed with a glass coverslip consisting of a round 3 mm glass coverslip (Warner Instruments; 64-0720 (CS-3R)) glued to a round 4 mm coverslip (Warner Instruments; 64-0724 (CS-4R)) with UV-cured optical glue (Norland Products; 7110). The coverslip was positioned with the 3 mm side placed directly over the cortical surface, while the 4 mm coverslip laid on the skull at the edges of the craniotomy. An instant adhesive (Loctite Instant Adhesive 495) was carefully dispensed along the edge of the 4 mm coverslip to secure it to the skull, taking care not to allow any spillover onto the brain. Lastly, the area around the cranial window was sealed with dental cement. Animals were given at least 3 weeks to recover and to ensure the window would be optically clear before imaging took place.

### Animal imaging

During imaging sessions, mice were anesthetized with ~30 µl chlorprothixene (1 mg ml$^{-1}$, intramuscular) and low levels of isoflurane (≤0.8%). Heart rate and respiration rate were continuously monitored to ensure consistent anesthetic depth and animals were kept on a heating pad to maintain body temperature. Either fluorescein isothiocyanate (FITC) dextran (MW = 2000 kDa) or Texas Red (70 kDa) dextran (5% [wt/vol] in saline) was injected retro-orbitally (20–40 µl) under an initial high level of isoflurane (>2%). When visual stimuli were presented, the injections were in the eye not being stimulated by the computer display monitor. Isoflurane levels were returned to ≤0.8% for at least 15 min before data collection began.

## Awake imaging

Animals were gradually acclimated to awake head restraint on a treadmill allowing for voluntary locomotion. Imaging experiments were conducted following a minimum of 2 weeks of training. Animals were briefly anesthetized with >2% isoflurane while the retro-orbital injection of the vascular dye was performed. We then allowed 15 min to recover before beginning the imaging.

### Equipment

Imaging data were collected using the Ultima 2P-Plus two-photon microscope system (Bruker Corporation – see *Figure 1C*). At MUSC we used a beta version of the Ultima 2P-Plus. The major differences from the full version at AU were that the beta version did not have the 594 nm LED for single-photon stimulation or the electro-tunable lens (ETL). At MUSC, an Insight X2 (Spectra-Physics, MKS Instruments Inc) was used for imaging and a FemtoTrain (Spectra-Physics: 1040 nm, ~3.5 W average power; <370 fs pulse width; >350 nJ pulse energy) was used for optogenetic stimulation. At AU we used the tunable line from the Insight X3 (Spectra-Physics) for imaging and the fixed 1045 nm line (~4 W average power; <170 fs pulse width; ~44 nJ pulse energy) from the same laser for stimulation. Dedicated fixed wavelength lasers used for two-photon optogenetics typically have lower repetition rates and higher pulse powers than imaging lasers like the Insight X3. Nevertheless, despite the lower pulse energy, the fixed line from the Insight X3 was sufficient to evoke constrictions – even deep in the cortex. No differences were observed in the results obtained at the two institutions and so results were combined.

Pockels Cells (350-80, Conoptics) were used to control laser power. The stimulation beam was passed through an SLM (HSP512-1064, Meadowlark Optics) to create three-dimensional activation patterns. A dichroic mirror (reflectance band from 1010 to 1070 nm) combined the two lasers downstream of the galvanometer mirrors. Imaging was performed using Nikon objectives (CFI75 LWD 16X W and CFI75 Apochromat 25XC W 1300).

Because of the intensity of the stimulation light (for both single- and two-photon illumination) and the potential for extremely bright fluorescence from excitation via the stimulation laser, the PMT detectors were fitted with fast mechanical shutters which block incoming light during the optical stimulation periods. This is seen in all the videos as a brief dimming of the image when the stimulation light is on. Our strategy was to use brief stimulation intervals at low repetition rates to allow sufficient data on arteriole diameter to be measured between stimulus pulses.

The LED on the Ultima 2P-Plus was positioned in the light path on the collection side of the primary dichroic (*Figure 1C*). To send this light to the objective, a cube was positioned in the path between the PMT and the objective containing both a mirror that reflects a band of light around the LED wavelength (570–605 nm) and a filter that blocks red light from reaching the PMT detectors (near-zero transmission band from 575 to 605 nm). This resulted in a modest reduction of the Texas Red signal in this mode.

### Optical stimulation
#### A. Single-photon

For single-photon stimulation, we used a 594 nm LED. We used 100ms pulses of light. When a train of pulses was given, there were 100 ms between pulses. Details of the pulse timing are given in the figure legends.

#### B. Two-photon

For two-photon stimulation, we used the SLM to split the stimulation laser beam into multiple spots (or beamlets). Each stimulation spot was spiraled over a small region (12–25 µm, five revolutions, one time per light pulse, see *Figure 4—figure supplement 1D*) using the galvanometer mirrors to increase the area of activation (*Packer et al., 2015*; *Carrillo-Reid et al., 2016*). Single 100 ms pulses were typically used. Laser powers were typically 50–200 mW per vessel, spread over four to six spots for pial arterioles, and two to three spots for penetrating arterioles. Although higher powers led to damage of vessel walls and dye leakage on the surface in pial arterioles, even the highest powers we used (>300 mW per vessel) in penetrating arterioles caused no visible damage or leakage. To determine the power of the stimulation spots, we used a power meter while presenting random patterns of four and six spots. It is not feasible to calibrate the power in each spot separately, and we therefore

divided the overall power by the number of spots to obtain an estimate of the power of each spot. The total power varied somewhat (typically <10%) from pattern to pattern and the amount of power each spot receives depends on its position on the SLM (**Mardinly et al., 2018**). Since it was not possible to know what spot pattern was required until we were positioning them on the vessels, the exact power applied to each vessel in each experiment is an estimate.

## Optical imaging

Images were usually collected using resonant scanning mode at 3.75 frames s$^{-1}$, except for **Figure 11— figure supplement 1** (15 frames s$^{-1}$). Vessels were imaged at 875 nm when labeled with Texas Red-dextran and 800 nm when imaged with FITC dextran.

Although 920 nm is in the tail of the activation spectrum for red-shifted opsins like ReaChR (**Chaigneau et al., 2016**), wavelengths in this range are typically used in all-optical (activation and imaging) neural approaches (**Packer et al., 2015**; **Carrillo-Reid et al., 2016**; **Marshel et al., 2019**). These previous studies reported that low laser powers for imaging prevents photo-activation from the imaging laser. We typically imaged Texas Red at 875 nm and saw no evidence that the imaging laser led to vasoconstriction. When we imaged at 920 nm, we did see slight constrictions in pial arterioles with moderate power levels (>20 mW), but under normal conditions, the vascular dye was sufficiently bright to reduce imaging power below these levels and eliminate constriction caused by the imaging laser. The possibility of constriction due to the imaging laser is important to keep in mind when choosing an opsin/indicator combination. Although more sensitive opsins are usually desirable because less light is needed to activate them and the stimulation beam can be split into more beamlets, it will also make the cells more readily activated by the imaging laser.

It is also important to consider the effects of the emitted fluorescence on the opsin. Texas Red emission overlaps the excitation peak for single-photon activation of ReaChR. This feature was elegantly used in a recent study as a method for spatially precise single-photon excitation (**Tong et al., 2020b**; **Tong et al., 2020a**). While we did not see evidence of a constant constriction during imaging caused by the emission of Texas Red (further constriction was always possible and vessels did not rapidly constrict at the start of an imaging run), the constriction seen with higher imaging powers that we attributed to imaging laser activation may have at least in part been caused by indirect actions of Texas Red emission on ReaChR.

## Visual stimuli

Drifting square-wave grating stimuli were presented on a 17-inch LCD monitor. The gratings were presented at 100% contrast, 30 cd m$^{-2}$ mean luminance, 1.5 Hz temporal frequency, and 0.04 cycles/ degree. The visual stimulus and the optogenetic pulses were both controlled by the imaging software, ensuring that the onset of both stimulation types were synchronized.

## Data analysis

Data were analyzed in Matlab (Mathworks) and ImageJ (National Institutes of Health). Blood vessel diameters were computed frame-by-frame using custom Matlab scripts (**O'Herron et al., 2016**; **O'Herron et al., 2020**; https://github.com/poherron/Vessel-Diameter-Code (copy archived at swh:1:rev:0d0dbe783ae4c9d61f5e40eee0b00814395586d7; **O'Herron, 2022**). Diameters were normalized by the average of 15 frames prior to stimulation onset. Repetitions were aligned by the onset of the optical stimulation when averaged together. The latency of constriction (**Figure 11— figure supplement 1**) was computed using the Curve Fitting toolbox in Matlab. A two-phase equation was fit to the diameter time-course data (see **O'Herron and von der Heydt, 2011**), where phase one was a constant (the baseline diameter) and phase two was an exponential curve (following the constriction phase of the response). The fit is given by:

$$A.\text{phase } one\,(t) + [A + C(exp(-(t-t_1)/\tau) - C)].\text{phase } two\,(t)$$

$$\text{where phase one }(t) = \begin{Bmatrix} 1\,for\,t < t_1 \\ 0\,else \end{Bmatrix} \text{ and phase two }(t) = \begin{Bmatrix} 1\,for\,t_1 \le t < 2\,\text{seconds} \\ 0\,\text{else} \end{Bmatrix}.$$

$A$ is a constant representing the baseline value (nearly 0), $C$ is the amplitude of the exponential, and $\tau$ is its time constant. The time point where the fit switches between the two phases is the parameter

$t_1$, which gives the onset latency. The data to fit are cut off at 2 s which was determined by eye to be near the rising phase of the response.

## Additional information

### Funding

| Funder | Grant reference number | Author |
|---|---|---|
| National Institute of Neurological Disorders and Stroke | NS110069 | Philip J O'Herron |
| National Institute on Aging | AG070507 | Philip J O'Herron |
| National Institute of Neurological Disorders and Stroke | NS106138 | Andy Y Shih |
| American Heart Association | 14GRNT20480366 | Andy Y Shih |
| National Center for Advancing Translational Sciences | UL1 TR001450 | David A Hartmann |
| National Center for Advancing Translational Sciences | TL1 TR001451 | David A Hartmann |
| National Institute of Neurological Disorders and Stroke | NS096868 | David A Hartmann |
| National Science Foundation | NSF1539034 | Prakash Kara |
| National Institute of Neurological Disorders and Stroke | NS097775 | Andy Y Shih |
| National Institute on Aging | AG062738 | Andy Y Shih |
| National Institute of Mental Health | MH111447 | Prakash Kara |

The funders had no role in study design, data collection, and interpretation, or the decision to submit the work for publication.

### Author contributions

Philip J O'Herron, Conceptualization, Resources, Data curation, Software, Formal analysis, Supervision, Funding acquisition, Validation, Investigation, Visualization, Methodology, Writing – original draft, Project administration, Writing – review and editing; David A Hartmann, Conceptualization, Data curation, Formal analysis, Funding acquisition, Investigation, Methodology, Writing – review and editing; Kun Xie, Investigation; Prakash Kara, Conceptualization, Resources, Software, Funding acquisition, Writing – review and editing; Andy Y Shih, Conceptualization, Resources, Supervision, Funding acquisition, Investigation, Visualization, Methodology, Project administration, Writing – review and editing

### Author ORCIDs

Philip J O'Herron  http://orcid.org/0000-0002-8137-9432
Prakash Kara  http://orcid.org/0000-0002-4285-1634

### Ethics

This study was performed in strict accordance with the recommendations in the Guide for the Care and Use of Laboratory Animals of the National Institutes of Health. All of the animals were handled according to approved Institutional Animal Care and Use Committee (IACUC) protocols of the Medical University of South Carolina and Augusta University (current protocol #0982).

Decision letter and Author response
Decision letter https://doi.org/10.7554/eLife.72802.sa1
Author response https://doi.org/10.7554/eLife.72802.sa2

## Additional files

### Supplementary files
• Transparent reporting form
• Source data 1. Individual vessel data used in the population plots.

### Data availability
All data included in this study are presented in the figures. The source data for the average data points presented in the paper are given in the Source Data file.

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
