## [Editor Report]

This paper will likely be of keen interest to researchers investigating vasculo-neuronal coupling – a proposed signaling mode opposite that of the more widely studied neuro-vascular coupling process. The optogenetics method described, inspired by methodology developed for interrogating ensembles of neurons, effectively enables simultaneous manipulation and monitoring of brain arteriole contractility in three dimensions.

---

## [Decision Letter]

**Decision letter after peer review:**

Thank you for submitting your article "Precise, 3-D optogenetic control of the diameter of single arterioles in vivo" for consideration by *eLife*. Your article has been reviewed by 3 peer reviewers, including Mark T Nelson as Reviewing Editor and Reviewer #1, and the evaluation has been overseen by Ronald Calabrese as the Senior Editor. The following individuals involved in review of your submission have agreed to reveal their identity: Anna Devor (Reviewer #2); Ravi Rungta (Reviewer #3).

Essential revisions:

1. The red-shifted opsin, ReaChR, represents an improvement over opsins used in previously described 3D neuronal activation/monitoring systems. In particular, brief single-photon stimulation (100 ms) of ReaChR led to rapid, robust arteriole constrictions throughout the activation volume, whereas a previous generation ChR2 opsin required stimulation for seconds to achieve slowly appearing constrictions.

2. Single-photon stimulation was capable of completing stopping blood flow in a "first order pre-capillary branch". (Not clear what is meant by the phrase "pre-capillary branch"; anatomically, penetrating arterioles feed capillary branches.) While this speaks to the effectiveness of the method, it also highlights potential supraphysiological effects of stimulation and the importance of titrating stimulus intensity/duration to achieve physiologically meaningful responses.

3. In assessing effects of laser power, the authors assert that "increasing the laser power only slightly expanded the range of constriction". This seems a bit of an overstatement, given that increasing power (30-fold) had a greater effect on the spread (3x) than the magnitude (2x) of the response.

4. The suggestion that penetrating brain arterioles possess a mechanism for upstream conduction of constrictive responses is intriguing (although this intrigue is tempered by the lack of experimental support for the operation of such a mechanism in the brain microvasculature).

5. The authors' premise for comparing contractile kinetics with sensory-evoked kinetics has issues. In attempting to use the kinetics of optogenetic-induced constriction to infer something about the kinetics of sensory-evoked dilation, they are implicitly assuming that the kinetics of contraction and dilation processes intrinsic to mural cells are the same. This is highlighted by their use of the phrase "kinetics of the vasculature", which elides the possibility that dilation and contraction kinetics intrinsic to mural cells are different. Support for this latter possibility is provided by a previous report on renal afferent arterioles showing that the kinetics of myogenic constriction in arterioles are "substantially faster" than those of dilation (PMID: 24173354). Thus, their data do not rule out the possibility that the delay between sensory stimulation and vascular response reflects a slower intrinsic dilatory response rather than the time course of neurovascular coupling mechanisms. Furthermore, arterioles have an internal elastic lamina (IEL), which also determines the rates and degree of constriction and dilation. The IEL ends with the arterioles, and vessels with ensheathing contractile pericytes (and downstream) lack the constraints of the IEL.

6. It's not at all clear how overriding sensory-evoked dilation with optogenetically generated constriction provides a means for distinguishing neural activity from vascular responses. In particular, it is not clear how performing this maneuver while monitoring neuronal activity can provide the suggested insight into "aspects" of functional hyperemia that are essential to neuronal function beyond the relatively trivial observation that there is a point at which blood flow is too low to support continued neuronal activity.

7. Presentation of high vs. low numerical aperture (NA) effects on X-Y and Z resolution is muddled. For high NA, the authors emphasize that the spread of constricting effects is greater in the Z plane than the X-Y plane. For low NA, they note "constrictions over a larger Z-range" (apparently compared to high NA but not clear), without indicating what the spread is in the X-Y plane. This leaves an apples-to-oranges comparison: greater spread in the Z plane compared with X-Y plane for low NA on the one hand versus greater spread in the Z plane with high NA compared with spread in the Z plane with low NA on the other. Need to show the same data for low and high NA (or make the rationale for the comparisons they do show clearer).

8. The authors write in very vague terms about potential applications of their methodology. They should make a greater effort to think through possible experimental applications and clearly present them.

9. Given the chronic nature of the optical window, it is not clear why imaging was done under anesthesia. This point requires explanation. There is a concern that targeting of the vessel wall not possible in awake animals due to brain motion. If yes, that would be a serious limitation of the methodology.

10. A major limitation of the technique is the poor axial point spread function of the SLM ReaChR activation. Although it is to be expected that the axial PSF is worst than the lateral PSF, the results are quite dramatic (with arteriole constriction being triggered when the SLM pattern is focused 150um from the arteriole, and at equivalent magnitude 200um away with the 0.8NA objective Figure 5-1). I think these experiments are important, but it would be helpful to provide some more control experiments to further characterize and help resolve the reason for this effect.

11. The bleach spots from their control experiment with the SLM focused 200um above or below the imaging plane, are not nearly the same size as the large axial PSF of the evoked constriction. One difference between the bleaching experiment and the SLM stimulation is that in the case of the SLM stimulation multiple spots are generated. Would it be possible to perform the bleaching control using the exact multi-spot pattern used for the experiments to ensure the multi spot pattern is not causing the SLM to generate a weird pattern in the z-plane?

12. Is it possible that there is still some 1-photon activation of ReaChR at 1040nm? This may be unlikely, but from the spectrum I found (Lin et al., 2013, PMID: 23995068), it was only tested up to 650 nm and the spectrum is quite broad with significant current evoked at all wavelengths tested.

13. If it is truly due to 2P activation generated within the cone of light, then this suggests that far higher power than necessary is being used (see work by Rickgauer and Tank – PMID: 19706471). In Figure 4 for example, the authors are able to trigger a nice local constriction with 5mW total spot power (>20 times less than is being used in the other experiments). If the same axial precision experiments are done with lower power does the constriction "PSF" decrease in width? Consistent with this idea, what is the result if the authors bypass the SLM and make a point scan 200um above the vessel at 115mW? Do they still trigger a constriction due to excitation with the cone of light? Finally, if the authors perform the experiment in Figure 3D (where they show nice xy precesion) and were then to move the focus of the SLM up in the z-plane, would they maintain this lateral specificity across the z-plane? It is important to properly characterize this axial "PSF" to establish power limitations for future studies.

14. Also as stated in the public review, although the authors state numbers of mice tested in the figure legends, the paper seems to be mostly composed of representative examples without quantification of the results across the other mice on which the method was tested. It is important to provide the average numbers and variability of all the experiments (either directly in the figures, or in the main text). Without this information, it is not possible to get a sense of reproducibility and variability.

15. The authors make comparisons between ReaChR and ChR2, although vascular dynamics are not directly compared between the 2 opsins using the same stimulation paradigm (e.g. line 94, This robust constriction (~20% from baseline level) to such a brief light stimulation is in stark contrast to activation with ChR2, where sustained stimulation over seconds was required for slow constrictions to appear (Hill et al., 2015; Tong et al., 2020a; Hartmann et al., 2021)), Although I appreciate that ReaChR may be preferable, the difference in kinetics of their vascular response is likely predominantly due to the nature of the stimulation (raster scanning vs. flood illumination or SLM) used here, rather than a difference in the opsin (ReaChR vs ChR2) as stated in this sentence. Supporting this thought, previous work has indeed shown rapid dilations and constrictions induced by activation of excitatory and inhibitory opsins with single photon epi-illumination (e.g. Abe et al., 2021, PMID: 34320360; Mateo et al., 2017, PMID: 29107517). The authors should modify the text appropriately. It would also be a nice (although not ultimately necessary) addition to compare their results with the SLM to raster / line scanning 2P activation of ReaChR on arterioles.

16. The mouse line used in this paper results in ReaChR expression in pericytes in addition to SMCs. The study would benefit from a brief description of what happens when they stimulate capillary pericytes with the SLM in comparison to their recently published results (Hartmann et al., 2021 – Nat Neurosci)?

---

## [Author Response]

Essential revisions:1. The red-shifted opsin, ReaChR, represents an improvement over opsins used in previously described 3D neuronal activation/monitoring systems. In particular, brief single-photon stimulation (100 ms) of ReaChR led to rapid, robust arteriole constrictions throughout the activation volume, whereas a previous generation ChR2 opsin required stimulation for seconds to achieve slowly appearing constrictions.

Thank you for pointing out this key takeaway from our manuscript. In Figure 9 of the revised manuscript, we provide a comparison of ReaChR-induced vasoconstriction, with data previously collected across microvascular zones using line-scanning in ChR2-expressing mice. These data show how ReaChR produces faster and more potent vasoconstriction in α-SMA expressing SMCs and ensheathing pericytes, but has similar effects on the slow contraction with capillary pericytes.

2. Single-photon stimulation was capable of completing stopping blood flow in a "first order pre-capillary branch". (Not clear what is meant by the phrase "pre-capillary branch"; anatomically, penetrating arterioles feed capillary branches.) While this speaks to the effectiveness of the method, it also highlights potential supraphysiological effects of stimulation and the importance of titrating stimulus intensity/duration to achieve physiologically meaningful responses.

We have removed the term “pre-capillary” to avoid causing confusion, and now use the term arteriole-capillary transition to denote the α-SMA positive segment that lies between the penetrating arteriole (0^th^ order) and the α-SMA low/negative capillaries (>4^th^ order). The rationale for this terminology is provided in our new review (Hartmann et al., 2022), which explains why the transitional zone should be considered a separate vessel type that is not arteriole and not capillary.

We agree with the reviewer that titration of stimulation power/duration will be important and will depend on the application. We addressed this point by performing measurements of arteriole diameter with graded laser powers (Figures 5 and 7). There are many parameters to explore, but for the purposes of this manuscript, we clarify that the effect is titratable and that users should define physiological ranges in their specific circumstances, which may differ based on the experimental goals, age of mice, arteriolar size and vascular zone, and other factors.

We also note that some applications may want to mimic pathophysiological levels of constriction, for example to mimic the effects of arterial vasospasm after subarachnoid hemorrhage, or ensheathing pericyte contraction with MCAo stroke (Hill et al., 2015), or to examine the neural consequences of transient small vessel occlusion.

3. In assessing effects of laser power, the authors assert that "increasing the laser power only slightly expanded the range of constriction". This seems a bit of an overstatement, given that increasing power (30-fold) had a greater effect on the spread (3x) than the magnitude (2x) of the response.

Thank you for pointing this out. We have re-worded this section to avoid the overstatement and to emphasize the results more clearly on the spatial spread of constriction relative to laser power.

The difference images in Figures 4B-C, G-H demonstrated that there was very limited spread of the constriction beyond the stimulation spots. We tested the effect of laser power on the spatial spread of constriction by stimulating with a broad range of power levels. We found that increasing the laser power led to a small increase in the spread of constriction. For example, a 30-fold increase in power (from 5 mW to 150 mW total power) led to ~3-fold increase in the spread of constriction (from ~25 µm to ~75 µm) (Figure 5A-H).

4. The suggestion that penetrating brain arterioles possess a mechanism for upstream conduction of constrictive responses is intriguing (although this intrigue is tempered by the lack of experimental support for the operation of such a mechanism in the brain microvasculature).

We are also intrigued by this hypothesis, which was supported by some evidence from a recent study of retinal vasculature. Kovacs-Oller *et al.,* showed using neurocytin tracer injections into capillary pericytes, that they are linked through gap junctions and there is upstream directional diffusion of tracer. Further, they showed that electrical stimulation of a pericyte could lead to directional constriction from capillaries back to the arteriole in the retina (Kovacs-Oller et al., 2020). The planar orientation of retinal vasculature makes this phenomenon easier to see. However, the 3D architecture of cortical vasculature is more challenging to study, particularly since the propagation along arterioles occurs along the Z axis, where spatiotemporal resolution of imaging is limited.

Given our new data on the effects of laser power on axial spread (see reply to points 10-13 below) and the difficulty in separating active propagation from out-of-focus activation, we think there is not sufficient evidence to claim that penetrating arterioles are propagating the signal through some active process. Further experiments, including studies of the mechanisms involved, will be needed to address this hypothesis. Therefore, we have removed any discussion of potential propagation of the signal, and instead focus on the relationship between laser power and axial resolution of activation.

5. The authors’ premise for comparing contractile kinetics with sensory-evoked kinetics has issues. In attempting to use the kinetics of optogenetic-induced constriction to infer something about the kinetics of sensory-evoked dilation, they are implicitly assuming that the kinetics of contraction and dilation processes intrinsic to mural cells are the same. This is highlighted by their use of the phrase “kinetics of the vasculature”, which elides the possibility that dilation and contraction kinetics intrinsic to mural cells are different. Support for this latter possibility is provided by a previous report on renal afferent arterioles showing that the kinetics of myogenic constriction in arterioles are “substantially faster” than those of dilation (PMID: 24173354). Thus, their data do not rule out the possibility that the delay between sensory stimulation and vascular response reflects a slower intrinsic dilatory response rather than the time course of neurovascular coupling mechanisms. Furthermore, arterioles have an internal elastic lamina (IEL), which also determines the rates and degree of constriction and dilation. The IEL ends with the arterioles, and vessels with ensheathing contractile pericytes (and downstream) lack the constraints of the IEL.

We thank the reviewer for this constructive critique. We agree that there are many issues in comparing kinetics between sensory evoked dilation and our optogenetic constriction. We have re-worded this section to avoid any mechanistic implications in the discussion of the kinetics of the different processes. However, we wish to still incorporate the details about the rapid kinetics of constriction to highlight the utility of the approach to intervene/perturb sensory-evoked responses, given that contraction can be titrated and precisely timed. We discuss the utility of this approach further below.

6. It’s not at all clear how overriding sensory-evoked dilation with optogenetically generated constriction provides a means for distinguishing neural activity from vascular responses. In particular, it is not clear how performing this maneuver while monitoring neuronal activity can provide the suggested insight into “aspects” of functional hyperemia that are essential to neuronal function beyond the relatively trivial observation that there is a point at which blood flow is too low to support continued neuronal activity.

Thank you for raising this point. We have added more detail to our thoughts on why over-riding functional hyperemia could provide insight into the dependence of neural activity on the blood flow increase. Neural circuits are extremely complex with many different sub-types of neurons playing different roles. These subtypes have been shown to have different metabolic sensitivities and thus, may be differentially affected by blocking functional hyperemia (Kann, 2016). This could lead to altered circuit activity which could have profound consequences for neural processing. Additionally, the energy budgets of different cellular functions within neurons are quite different (Howarth et al., 2012) and reducing available energy by blocking functional hyperemia could lead to differing degrees of dysfunction across important cellular processes (e.g. re-establishing the membrane potential, recycling neurotransmitters) which could again have important consequences for neural coding. Furthermore, it has been shown that there is a steep gradient of oxygen moving away from penetrating arterioles, and so neurons at greater distances from vessels may be differentially affected by blocking the hyperemic response (Devor et al., 2011).

7. Presentation of high vs. low numerical aperture (NA) effects on X-Y and Z resolution is muddled. For high NA, the authors emphasize that the spread of constricting effects is greater in the Z plane than the X-Y plane. For low NA, they note “constrictions over a larger Z-range” (apparently compared to high NA but not clear), without indicating what the spread is in the X-Y plane. This leaves an apples-to-oranges comparison: greater spread in the Z plane compared with X-Y plane for low NA on the one hand versus greater spread in the Z plane with high NA compared with spread in the Z plane with“low NA on the other. Need to show the same data for low and high NA (or make the rationale for the comparisons they do show clearer).

We thank the reviewer for this comment. Please see our response to comment 10 below, which brings up a similar concern.

8. The authors write in very vague terms about potential applications of their methodology. They should make a greater effort to think through possible experimental applications and clearly present them.

In addition to our response above on the utility of over-riding arteriole dilation during functional hyperemia, we have added to the discussion more potential uses of the technique. These include: (1) To be able to manipulate blood flow without using pharmacology or having to induce neural activity could be useful for a variety of studies involving intrinsic reactivity and compliance of vessels in both health and disease. (2) The different microvascular zones have distinct contractile kinetics. There are details that remain unstudied, such as the kinetics of different sized vessels, their location in the network, their identity as collateral arterioles or pial arterioles. Vascular optogenetics can dissect the contractile characteristics of different vessel types, similar to probing a circuit board. (3) Studies of the physiological significance of vasomotion, with respect to brain clearance of metabolic waste products. Being able to directly drive vasomotion and alter its amplitude and frequency will be an important tool for studies in this field. (4) Functional hyperemia is also impaired in many diseases, but this dysfunction could arise from impaired activity of neurons, astrocytes, or vessels. Therefore, a method to disentangle specific changes to blood vessels in vivo could be useful for understanding the vascular contributions to such diseases.

9. Given the chronic nature of the optical window, it is not clear why imaging was done under anesthesia. This point requires explanation. There is a concern that targeting of the vessel wall not possible in awake animals due to brain motion. If yes, that would be a serious limitation of the methodology.

To ensure that our method is compatible with awake experiments, we have added awake data to the manuscript (Figure 10). We show that individual vessels can be independently targeted in the awake animal and the outcomes are not profoundly different than in the anesthetized state. As with all awake experiments, due diligence must be taken to ensure the preparation is as stable as possible, and the occasional trial may have to be removed if motion artifacts are too large.

10. A major limitation of the technique is the poor axial point spread function of the SLM ReaChR activation. Although it is to be expected that the axial PSF is worst than the lateral PSF, the results are quite dramatic (with arteriole constriction being triggered when the SLM pattern is focused 150um from the arteriole, and at equivalent magnitude 200um away with the 0.8NA objective Figure 5-1). I think these experiments are important, but it would be helpful to provide some more control experiments to further characterize and help resolve the reason for this effect.

We agree with the reviewers on this point. We conducted several new experiments to help elucidate the limits of axial resolution. First, we have removed the comparison between objectives with different numerical apertures. This leads to unnecessary confusion, and it is common knowledge that lower NA objectives will have poorer resolution in the axial plane. We now mention this as a factor to consider but have removed it from the figures. Second, we have shown, as the reviewer suggests below, that the stimulation power used has a dramatic effect on the axial spread of constriction (Figure 6E and Figure 7). Low powers indeed show a narrower axial spread. However, we typically use higher powers (near or above 100 mW) to generate large constrictions in penetrating arterioles, and we also include these levels to show the greater axial spread they cause. In summary, we confirm with lower powers that 3D precision can be achieved with the two-photon optogenetic technique, and we show that higher powers can be used to broadly constrict penetrating arterioles for studies seeking to modulate blood flow in columns of cortical tissue supplied by penetrating arterioles.

11. The bleach spots from their control experiment with the SLM focused 200um above or below the imaging plane, are not nearly the same size as the large axial PSF of the evoked constriction. One difference between the bleaching experiment and the SLM stimulation is that in the case of the SLM stimulation multiple spots are generated. Would it be possible to perform the bleaching control using the exact multi-spot pattern used for the experiments to ensure the multi spot pattern is not causing the SLM to generate a weird pattern in the z-plane?

The bleached spots were indeed multiple spots, as used with the vessels. We only displayed one to see the XZ projection, but there were actually 3 burned at a time. More importantly, bleaching spots on a slide is different from activating the opsin. The slide is optimally fluorescent at wavelengths ~800 nm and so bleaching with 1040 nm takes much more power and time (250-350 mW for 10-20 seconds) than opening highly light-sensitive channels at their optimal activation wavelengths. This is most likely the reason for the relatively small span of bleaching relative to opsin activation.

12. Is it possible that there is still some 1-photon activation of ReaChR at 1040nm? This may be unlikely, but from the spectrum I found (Lin et al., 2013, PMID: 23995068), it was only tested up to 650 nm and the spectrum is quite broad with significant current evoked at all wavelengths tested.

Please see our response to comment 13 below, which addresses comment 12 and 13 together.

13. If it is truly due to 2P activation generated within the cone of light, then this suggests that far higher power than necessary is being used (see work by Rickgauer and Tank – PMID: 19706471). In Figure 4 for example, the authors are able to trigger a nice local constriction with 5mW total spot power (>20 times less than is being used in the other experiments). If the same axial precision experiments are done with lower power does the constriction "PSF" decrease in width? Consistent with this idea, what is the result if the authors bypass the SLM and make a point scan 200um above the vessel at 115mW? Do they still trigger a constriction due to excitation with the cone of light? Finally, if the authors perform the experiment in Figure 3D (where they show nice xy precesion) and were then to move the focus of the SLM up in the z-plane, would they maintain this lateral specificity across the z-plane? It is important to properly characterize this axial “PSF” to establish power limitations for future studies.

Our reply to comment #10 above and the experiments/data that were added to address that are also relevant to this question.

Adding to that, we think that, given the new data with lower powers (Figure 6E,F), the more likely explanation is that this activation is a 2P rather than a 1P process. The data shows that with much lower powers we can still achieve robust constriction (in pial arterioles) in the stimulation plane but very little 50 µm deeper, indicating that the cone of light above the focal point is insufficient to trigger optogenetic activation.

Constricting penetrating arterioles hundreds of microns deep in the tissue requires more laser power (see Figure 7), but again, we show that there is a power range where constriction at the stimulation plane is attainable with good axial resolution. This relation between laser power and axial resolution is now much clearer in the manuscript.

We have also added new data to address the question about the lateral resolution of stimulation as the SLM is moved away from the imaging plane. Indeed, outside the focal plane of the stimulation, the activation is greater away from the vessel wall then when in the focal plane (compare the red bar to the orange and yellow bars in Figure 6F). However, the constrictions are quite minimal at all depth planes when there is just a small lateral distance between the stimulation spots and the vessel wall (Figure 6F, right panel). Nonetheless, as the 200 mW condition shows, with high enough powers, the cone of light can still activate over a wide region.

We think it is unlikely that distortion from the SLM is the cause of the axial resolution. As just mentioned, we show that with lower powers, one can achieve much higher resolution. Also, the photobleaching of spots on a slide shows that the SLM is positioning the beam accurately (Figure 6-1). We sought to do this suggested experiment of bypassing the SLM, but there are several limitations. First, there is no easy way to image 200 µm out of the focal plane without the SLM. An electrotunable lens (ETL) allows us to jump rapidly in Z, but short of mis-calibrating the system, we are limited to around 100 µm Z-span. There are other technical limitations which might make this hard to interpret, such as synchronizing the ETL position with the timing of the light stimulation and the detector shutter to ensure that light is only presented while the objective is focused in the targeted plane. Thus, we hope that our data showing the axial resolution at different powers is sufficient to answer the reviewer’s query.

14. Also as stated in the public review, although the authors state numbers of mice tested in the figure legends, the paper seems to be mostly composed of representative examples without quantification of the results across the other mice on which the method was tested. It is important to provide the average numbers and variability of all the experiments (either directly in the figures, or in the main text). Without this information, it is not possible to get a sense of reproducibility and variability.

We agree that showing population averages will be more informative to the field. In the original submission, we showed mostly examples because the large parameter space (size and number of spots, position on vessels, duration and intensity of stimulation; if a stimulation train, the duration, number, and inter-pulse interval of stimulation) was explored in the early data rather than picking one set of conditions. However, we have now collected new data where parameters were typically the same and included population average plots in the figures that previously had only individual examples (Figures 2G,I, 4I,M, 4-1C, 5I, 6E,F, 7, 11-2 ) as well as the new data (Figures 8, 9, 10).

15. The authors make comparisons between ReaChR and ChR2, although vascular dynamics are not directly compared between the 2 opsins using the same stimulation paradigm (e.g. line 94, This robust constriction (~20% from baseline level) to such a brief light stimulation is in stark contrast to activation with ChR2, where sustained stimulation over seconds was required for slow constrictions to appear (Hill et al., 2015; Tong et al., 2020a; Hartmann et al., 2021)), Although I appreciate that ReaChR may be preferable, the difference in kinetics of their vascular response is likely predominantly due to the nature of the stimulation (raster scanning vs. flood illumination or SLM) used here, rather than a difference in the opsin (ReaChR vs ChR2) as stated in this sentence. Supporting this thought, previous work has indeed shown rapid dilations and constrictions induced by activation of excitatory and inhibitory opsins with single photon epi-illumination (e.g. Abe et al., 2021, PMID: 34320360; Mateo et al., 2017, PMID: 29107517). The authors should modify the text appropriately. It would also be a nice (although not ultimately necessary) addition to compare their results with the SLM to raster / line scanning 2P activation of ReaChR on arterioles.

Thank you for this excellent question. We have addressed comments 15 and 16 together below since they are related.

16. The mouse line used in this paper results in ReaChR expression in pericytes in addition to SMCs. The study would benefit from a brief description of what happens when they stimulate capillary pericytes with the SLM in comparison to their recently published results (Hartmann et al., 2021 – Nat Neurosci)?

We conducted new experiments stimulating pericytes of the capillary zone and arteriole-capillary transition (ACT) zone (Figure 8), and added new analysis comparing this data with the ChR2 data from Hartmann et al., 2021 to address these points (Figure 9). We also compared SLM stimulation of various durations versus continuous line-scanning across the vessel lumen, the latter approach being what was used in Hartmann et al., 2021. The time course of constriction to line-scanning stimulation with ReaChR is similar to that previously observed, with penetrating arterioles showing fast and robust contraction, while ACT vessels and capillaries had slow and steady constrictions, though ACT vessels showed stronger constrictions of the two. However, single or continuous SLM pulses showed robust penetrating arteriole constriction and weak constriction of ACT vessels, but no constriction in capillary vessels. This difference between line scan and SLM stimulation with ACT vessels and capillaries is likely due to the intermittent SLM stimulation versus the virtually constant line scan stimulation, and we discuss this in the revised manuscript.

Our analyses comparing ReaChR and ChR2 responses to line scan stimulation revealed that ReaChR constricts SMA positive vessels (penetrating arterioles and ACT vessels) more rapidly than ChR2. However, capillaries showed the same kinetics of constriction between the two opsins. Thus, although both ReaChR and ChR2 can be used to stimulate mural cells across microvascular zones, ReaChR generates more rapid constrictions with α-SMA positive vessels. Further, the stimulation approach (line scanning vs intermittent pulses) matters for constriction of capillaries.

References

Devor A, Sakadžić S, Saisan PA, Yaseen MA, Roussakis E, Srinivasan VJ, Vinogradov SA, Rosen BR, Buxton RB, Dale AM, Boas DA (2011) “Overshoot” of O2 Is Required to Maintain Baseline Tissue Oxygenation at Locations Distal to Blood Vessels. The Journal of Neuroscience 31:13676-13681.

Hartmann DA, Coelho-Santos V, Shih AY (2022) Pericyte Control of Blood Flow Across Microvascular Zones in the Central Nervous System. Annu Rev Physiol 84:331-354.

Hill RA, Tong L, Yuan P, Murikinati S, Gupta S, Grutzendler J (2015) Regional Blood Flow in the Normal and Ischemic Brain Is Controlled by Arteriolar Smooth Muscle Cell Contractility and Not by Capillary Pericytes. Neuron 87:95-110.

Howarth C, Gleeson P, Attwell D (2012) Updated energy budgets for neural computation in the neocortex and cerebellum. J Cereb Blood Flow Metab 32:1222-1232.

Kann O (2016) The interneuron energy hypothesis: Implications for brain disease. Neurobiol Dis 90:75-85.

Kovacs-Oller T, Ivanova E, Bianchimano P, Sagdullaev BT (2020) The pericyte connectome: spatial precision of neurovascular coupling is driven by selective connectivity maps of pericytes and endothelial cells and is disrupted in diabetes. Cell Discov 6:39-39.

Mateo C, Knutsen PM, Tsai PS, Shih AY, Kleinfeld D (2017) Entrainment of Arteriole Vasomotor Fluctuations by Neural Activity Is a Basis of Blood-Oxygenation-Level-Dependent "Resting-State" Connectivity. Neuron 96:936-948.e933.